# Sensory processing in humans and mice fluctuates between external and internal modes

**Veith Weilnhammer** [1,2,3] *, **Heiner Stuke**[1,2], **Kai Standvoss**[1], **Philipp Sterzer**[4]

**1** Department of Psychiatry, Charité-Universitätsmedizin Berlin, corporate member of Freie Universität Berlin and Humboldt-Universität zu Berlin, Berlin, Germany, **2** Berlin Institute of Health, Charité-Universitätsmedizin Berlin and Max Delbrück Center, Berlin, Germany, **3** Helen Wills Neuroscience Institute, University of California Berkeley, Berkeley, California, United States of America, **4** Department of Psychiatry (UPK), University of Basel, Basel, Switzerland

* veith.weilnhammer@gmail.com

**Data Availability Statement:** All materials associated with this submission are available on an accompanying Github repository (https://github.com/veithweilnhammer/Modes, DOI: https://

## Abstract

Perception is known to cycle through periods of enhanced and reduced sensitivity to external information. Here, we asked whether such slow fluctuations arise as a noise-related epiphenomenon of limited processing capacity or, alternatively, represent a structured mechanism of perceptual inference. Using 2 large-scale datasets, we found that humans and mice alternate between externally and internally oriented modes of sensory analysis. During external mode, perception aligns more closely with the external sensory information, whereas internal mode is characterized by enhanced biases toward perceptual history. Computational modeling indicated that dynamic changes in mode are enabled by 2 interlinked factors: (i) the integration of subsequent inputs over time and (ii) slow antiphase oscillations in the impact of external sensory information versus internal predictions that are provided by perceptual history. We propose that between-mode fluctuations generate unambiguous error signals that enable optimal inference in volatile environments.

## 1. Introduction

The capacity to respond to changes in the environment is a defining feature of life [1–3]. Intriguingly, the ability of living things to process their surroundings fluctuates considerably over time [4,5]. In humans and mice, perception [6–12], cognition [13], and memory [14] cycle through prolonged periods of enhanced and reduced sensitivity to external information, suggesting that the brain detaches from the world in recurring intervals that last from milliseconds to seconds and even minutes [4]. Yet, breaking from external information is risky, as swift responses to the environment are often crucial to survival.

What could be the reason for these fluctuations in perceptual performance [11]? First, periodic fluctuations in the ability to parse external information [11,15,16] may arise simply due to bandwidth limitations and noise. Second, it may be advantageous to actively reduce the costs of neural processing by seeking sensory information only in recurring intervals [17], otherwise

zenodo.org/records/10019948). We included all relevant data and code for the generation of the manuscript in the R-Markdown format.

**Funding:** VW was funded by the Leopoldina Academy of Sciences (grant number: LDPS2022-16, https://www.leopoldina.org/en/leopoldina-home/) and the German Research Foundation DFG (grant number: STE 1430/8-1, https://www.dfg.de) VW and HS were funded by the Berlin Institute of Health Clinician Scientist Program (https://www.bihealth.org/en/translation/innovation-enabler/academy/bih-charite-clinician-scientist-program). PS was funded by the German Research Foundation DFG (grant number: STE 1430/8-1, https://www.dfg.de) and the German Ministry for Research and Education (ERA-NET NEURON program, grant number: 01EW2007A, https://www.neuron-eranet.eu/). The funders had no role in study design, data collection and analysis, decision to publish, or preparation of the manuscript.

**Competing interests:** The authors have declared that no competing interests exist.

**Abbreviations:** AIC, Akaike information criterion; IBL, International Brain Laboratory; MAD, median absolute distance; RT, response time; SD, standard deviation; TD, trial duration.

relying on random or stereotypical responses to the external world. Third, spending time away from the ongoing stream of sensory inputs may also reflect a functional strategy that facilitates flexible behavior and learning [18]: Intermittently relying more strongly on information acquired from past experiences may enable agents to build up stable internal predictions about the environment despite an ongoing stream of external sensory signals [19]. By the same token, recurring intervals of enhanced sensitivity to external information may help to detect changes in both the state of the environment and the amount of noise that is inherent in sensory encoding [19].

In this work, we sought to elucidate whether periodicities in the sensitivity to external information represent an epiphenomenon of limited processing capacity or, alternatively, result from a structured and adaptive mechanism of perceptual inference. To this end, we analyzed 2 large-scale datasets on perceptual decision-making in humans [20] and mice [21]. When less sensitive to external stimulus information, humans and mice did not behave more randomly but showed stronger serial dependencies in their perceptual choices [22–33]. These serial dependencies may be understood as driven by internal predictions that reflect the autocorrelation of natural environments [34] and bias perception toward preceding experiences [30,31,35]. Computational modeling indicated that ongoing changes in perceptual performance may be driven by systematic fluctuations between externally and internally oriented *modes* of sensory analysis. We suggest that such *bimodal inference* may help to build stable internal representations of the sensory environment despite an ongoing stream of sensory information.

## 2. Results

### 2.1 Human perception fluctuates between epochs of enhanced and reduced sensitivity to external information

We began by selecting 66 studies from the Confidence database [20] that investigated how human participants ($N$ = 4,317) perform binary perceptual decisions (Fig 1A; see Methods for details on inclusion criteria). As a metric for perceptual performance (i.e., the sensitivity to external sensory information), we asked whether the participant's response and the presented stimulus matched (*stimulus-congruent* choices) or differed from each other (*stimulus-incongruent* choices; Fig 1B and 1C) in a total of 21.05 million trials.

In a first step, we asked whether the ability to accurately perceive sensory stimuli is constant over time or, alternatively, fluctuates in periods of enhanced and reduced sensitivity to external information. We found perception to be stimulus-congruent in 73.46% ± 0.15% of trials (mean ± standard error of the mean; Fig 2A), which was highly consistent across the selected studies (S1A Fig). In line with previous work [8], we found that the probability of stimulus-congruence was not independent across successive trials: At the group level, stimulus-congruent perceptual choices were significantly autocorrelated for up to 15 trials (Fig 2B), controlling for task difficulty and the sequence of presented stimuli (S2 Fig).

At the level of individual participants, the autocorrelation of stimulus-congruence exceeded the respective autocorrelation of randomly permuted data within an interval of $3.24 ± 2.39 \times 10^{-3}$ trials (Fig 2C). In other words, if a participant's experience was congruent (or incongruent) with the external stimulus information at a given trial, her perception was more likely to remain stimulus-congruent (or stimulus-incongruent) for approximately 3 trials into the future. The autocorrelation of stimulus-congruence was corroborated by logistic regression models that successfully predicted the stimulus-congruence of perception at the index trial $t$ = 0 from the stimulus-congruence at the preceding trials within a lag of 16 trials (S3 Fig).

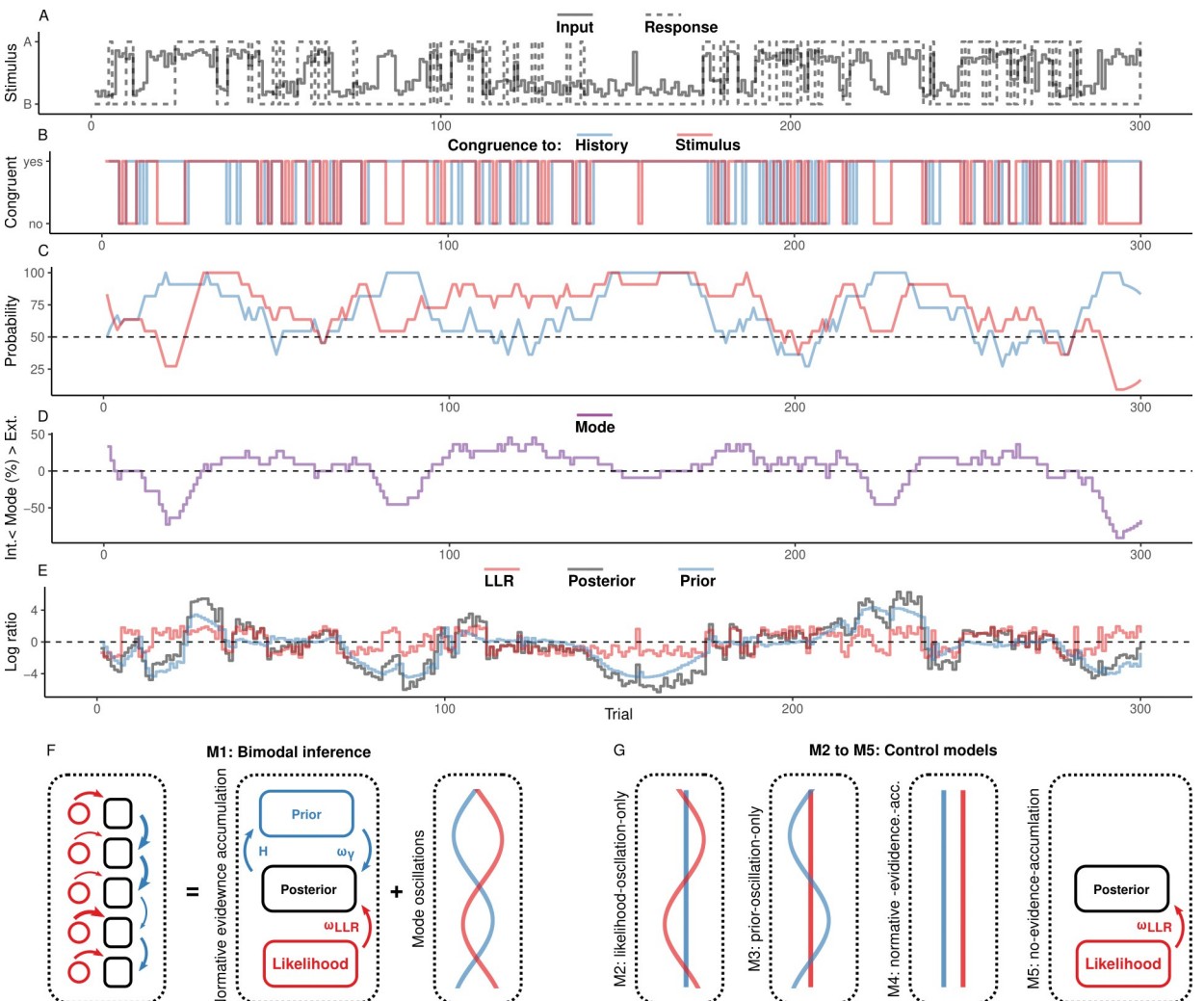

**Fig 1. Concept.** (**A**) In binary perceptual decision-making, a participant is presented with stimuli from 2 categories (A vs. B; solid line) and reports consecutive perceptual choices via button presses (solid line). All panels below refer to these stimulated example data. (**B**) When the response matches the external stimulus information (i.e., overlap between the dotted and solid lines in panel (**A**)), perceptual choices are *stimulus-congruent* (red line). When the response matches the response at the preceding trial, perceptual choices are *history-congruent* (blue line). (**C**) The dynamic probabilities of stimulus- and history-congruence (i.e., computed in sliding windows of ±5 trials) fluctuate over time. (**D**) The *mode* of perceptual processing is derived by computing the difference between the dynamic probabilities of stimulus- and history-congruence. Values above 0% indicate a bias toward external information, whereas values below 0% indicate a bias toward internal information. (**E**) In computational modeling, internal mode is caused by an enhanced impact of perceptual history. This causes the posterior (black line) to be close to the prior (blue line). Conversely, during external mode, the posterior is close to the sensory information (log likelihood ratio, red line). (**F**) The bimodal inference model (M1) explains fluctuations between externally and internally biased modes (left panel) by 2 interacting factors: a normative accumulation of evidence according to parameter $H$ (middle panel), and antiphase oscillations in the precision terms $\omega_{LLR}$ and $\omega_{\psi}$ (right panel). (**G**) The control models M2-M5 were constructed by successively removing the antiphase oscillations and the integration of information from the bimodal inference model. Please note that the normative-evidence-accumulation model (M4) corresponds to the model proposed by Glaze and colleagues [51]. In the no-evidence-accumulation model (M5), perceptual decisions depend only on likelihood information (flat priors).

These results confirm that the ability to process sensory signals is not constant over time but unfolds in multitrial epochs of enhanced and reduced sensitivity to external information [8]. As a consequence of this autocorrelation, the dynamic probability of stimulus-congruent perception (i.e., computed in sliding windows of ±5 trials; Fig 1C) fluctuated considerably within participants (average minimum: 35.46% ± 0.22%, maximum: 98.27% ± 0.07%). In line with previous findings [9], such fluctuations in the sensitivity to external information had a

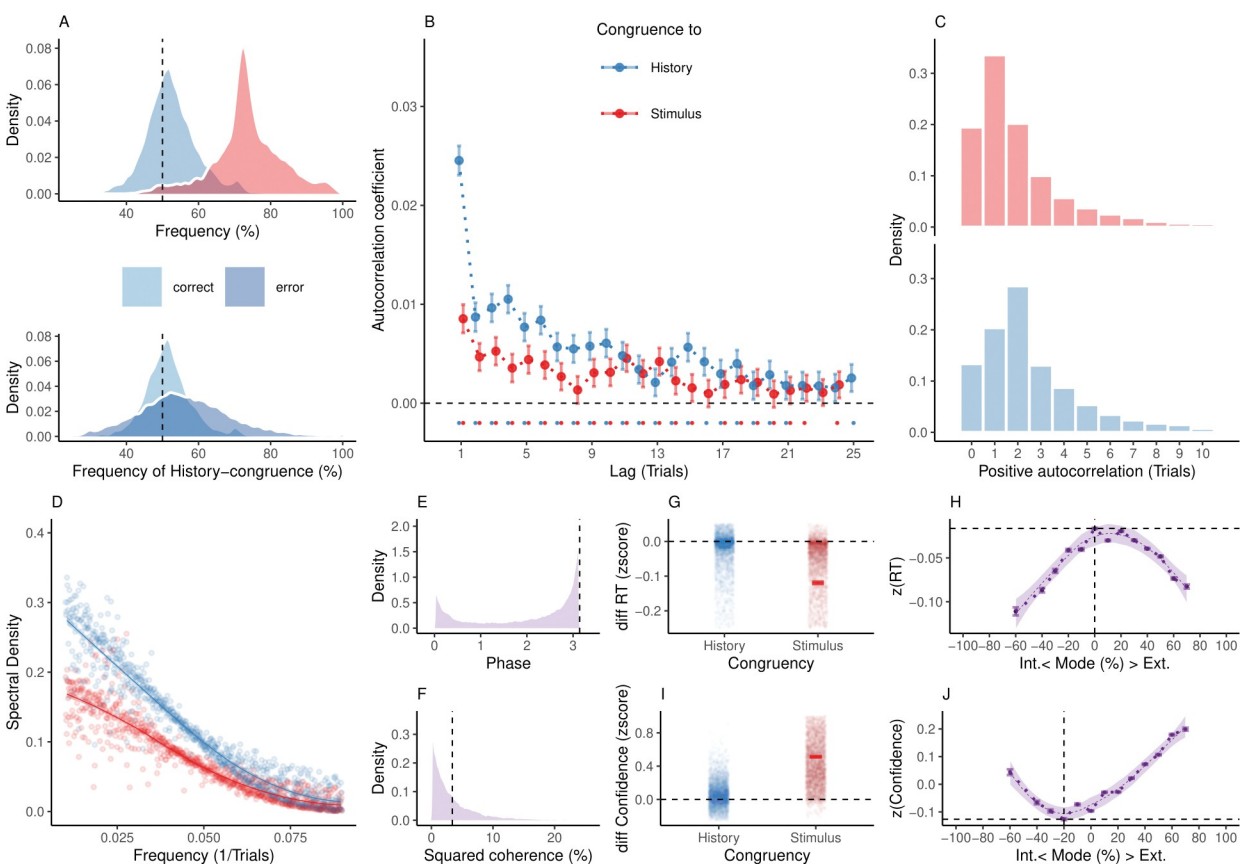

**Fig 2. Internal and external modes in human perceptual decision-making. (A)** In humans, perception was stimulus-congruent in 73.46% ± 0.15% (in red) and history-congruent in 52.7% ± 0.12% of trials (in blue; upper panel). History-congruent perceptual choices were more frequent when perception was stimulus-incongruent (i.e., on *error* trials; lower panel), indicating that history effects impair performance in randomized psychophysical designs. **(B)** Relative to randomly permuted data, we found highly significant autocorrelations of stimulus-congruence and history-congruence (dots indicate intercepts $\neq 0$ in trial-wise linear mixed effects modeling at $p < 0.05$). Across trials, the autocorrelation coefficients were best fit by an exponential function (adjusted $R^2$ for stimulus-congruence: 0.53; history-congruence: 0.72) as compared to a linear function (adjusted $R^2$ for stimulus-congruence: 0.53; history-congruence: 0.51), decaying at a rate of $\gamma = -1.92 \times 10^{-3} \pm 4.5 \times 10^{-4}$ (T($6.88 \times 10^4$) = $-4.27$, $p = 1.98 \times 10^{-5}$) for stimulus-congruence and at a rate of $\gamma = -6.11 \times 10^{-3} \pm 5.69 \times 10^{-4}$ (T($6.75 \times 10^4$) = $-10.74$, $p = 7.18 \times 10^{-27}$) for history-congruence. **(C)** Here, we depict the number of consecutive trials at which autocorrelation coefficients exceeded the respective autocorrelation of randomly permuted data within individual participants. For stimulus-congruence (upper panel), the lag of positive autocorrelation amounted to $3.24 \pm 2.39 \times 10^{-3}$ on average, showing a peak at trial t+1 after the index trial. For history-congruence (lower panel), the lag of positive autocorrelation amounted to $4.87 \pm 3.36 \times 10^{-3}$ on average, peaking at trial t+2 after the index trial. **(D)** The smoothed probabilities of stimulus- and history-congruence (sliding windows of ±5 trials) fluctuated as a scale-invariant process with a 1/f power law, i.e., at power densities that were inversely proportional to the frequency. **(E)** The distribution of phase shift between fluctuations in stimulus- and history-congruence peaked at half a cycle ($\pi$ denoted by dotted line). **(F)** The average squared coherence between fluctuations in stimulus- and history-congruence (dotted line) amounted to $6.49 \pm 2.07 \times 10^{-3}$%. **(G)** We observed faster RTs for both stimulus-congruence (as opposed to stimulus-incongruence, $\beta = -0.14 \pm 1.6 \times 10^{-3}$, T($1.99 \times 10^6$) = $-85.84$, $p < 2.2 \times 10^{-308}$) and history-congruence ($\beta = -9.56 \times 10^{-3} \pm 1.37 \times 10^{-3}$, T($1.98 \times 10^6$) = $-6.97$, $p = 3.15 \times 10^{-12}$). **(H)** The mode of perceptual processing (i.e., the difference between the smoothed probability of stimulus- vs. history-congruence) showed a quadratic relationship to RTs, with faster RTs for stronger biases toward both external sensory information and internal predictions provided by perceptual history ($\beta_2 = -19.86 \pm 0.52$, T($1.98 \times 10^6$) = $-38.43$, $p = 5 \times 10^{-323}$). The horizontal and vertical dotted lines indicate maximum RT and the associated mode, respectively. **(I)** Confidence was enhanced for both stimulus-congruence (as opposed to stimulus-incongruence, $\beta = 0.48 \pm 1.38 \times 10^{-3}$, T($2.06 \times 10^6$) = 351.54, $p < 2.2 \times 10^{-308}$) and history-congruence ($\beta = 0.04 \pm 1.18 \times 10^{-3}$, T($2.06 \times 10^6$) = 36.85, $p = 3.25 \times 10^{-297}$). **(J)** In analogy to RTs, we found a quadratic relationship between the mode of perceptual processing and confidence, which increased when both externally and internally biased modes grew stronger ($\beta_2 = 39.3 \pm 0.94$, T($2.06 \times 10^6$) = 41.95, $p < 2.2 \times 10^{-308}$). The horizontal and vertical dotted lines indicate minimum confidence and the associated mode, respectively.

power density that was inversely proportional to the frequency in the slow spectrum [11] (power $\sim 1/f^\beta$, $\beta = -1.32 \pm 3.14 \times 10^{-3}$, T($1.84 \times 10^5$) = $-419.48$, $p < 2.2 \times 10^{-308}$; Fig 2D). This feature, which is also known as a *1/f power law* [36,37], represents a characteristic of scale-free

fluctuations in complex dynamic systems such as the brain [38] and the cognitive processes it entertains [9,10,13,39,40].

## 2.2 Humans fluctuate between external and internal modes of sensory processing

In a second step, we sought to explain why perception cycles through periods of enhanced and reduced sensitivity to external information [4]. We reasoned that observers may intermittently rely more strongly on internal information, i.e., on predictions about the environment that are constructed from previous experiences [19,31].

In perception, *serial dependencies* represent one of the most basic internal predictions that cause perceptual decisions to be systematically biased toward preceding choices [22–33]. Such effects of perceptual history mirror the continuity of the external world, in which the recent past often predicts the near future [30,31,34,35,41]. Therefore, as a metric for the perceptual impact of internal information, we computed whether the participant's response at a given trial matched or differed from her response at the preceding trial (*history-congruent* and *history-incongruent perception*, respectively; Fig 1B and 1C).

First, we confirmed that perceptual history played a significant role in perception despite the ongoing stream of external information. With a global average of 52.7% ± 0.12% history-congruent trials, we found a small but highly significant perceptual bias towards preceding experiences ($\beta = 16.18 \pm 1.07$, $T(1.09 \times 10^3) = 15.07$, $p = 10^{-46}$; Fig 2A) that was largely consistent across studies (S1B Fig) and more pronounced in participants who were less sensitive to external sensory information (S1C Fig). Importantly, history-congruence was not a corollary of the sequence of presented stimuli: History-congruent perceptual choices were more frequent at trials when perception was stimulus-incongruent (56.03% ± 0.2%) as opposed to stimulus-congruent (51.77% ± 0.11%, $\beta = -4.26 \pm 0.21$, $T(8.57 \times 10^3) = -20.36$, $p = 5.28 \times 10^{-90}$; Fig 2A, lower panel). Despite being adaptive in autocorrelated real-world environments [19,34,35,42], perceptual history thus represented a source of bias in the randomized experimental designs studied here [24,28,30,31,43]. These serial biases were effects of choice history, i.e., driven by the experiences reported at the preceding trial, and could not be attributed to stimulus history, i.e., to effects of the stimuli presented at the preceding trial (S1 Text).

Second, we asked whether perception cycles through multitrial epochs during which perception is characterized by stronger or weaker biases toward preceding experiences. In close analogy to stimulus-congruence, we found history-congruence to be significantly autocorrelated for up to 21 trials (Fig 2B), while controlling for task difficulty and the sequence of presented stimuli (S2 Fig). In individual participants, the autocorrelation of history-congruence was elevated above randomly permuted data for a lag of 4.87 ± 3.36×10⁻³ trials (Fig 2C), confirming that the autocorrelation of history-congruence was not only a group-level phenomenon. The autocorrelation of history-congruence was corroborated by logistic regression models that successfully predicted the history-congruence of perception at an index trial $t = 0$ from the history-congruence at the preceding trials within a lag of 17 trials (S3 Fig).

Third, we asked whether the impact of internal information fluctuates as a scale-invariant process with a 1/f power law (i.e., the feature typically associated with fluctuations in the sensitivity to external information [9,10,13,39,40]). The dynamic probability of history-congruent perception (i.e., computed in sliding windows of ±5 trials; Fig 1C) varied considerably over time, ranging between a minimum of 12.77% ± 0.14% and a maximum 92.23% ± 0.14%. In analogy to stimulus-congruence, we found that history-congruence fluctuated at power densities that were inversely proportional to the frequency in the slow spectrum [11] (power $\sim 1/f^\beta$, $\beta = -1.34 \pm 3.16 \times 10^{-3}$, $T(1.84 \times 10^5) = -423.91$, $p < 2.2 \times 10^{-308}$; Fig 2D).

Finally, we ensured that fluctuations in stimulus- and history-congruence are linked to each other. When perceptual choices were less biased toward external information, participants relied more strongly on internal information acquired from perceptual history (and vice versa, $\beta = -0.05 \pm 5.63 \times 10^{-4}$, T($2.1 \times 10^{6}$) = −84.21, p < $2.2 \times 10^{-308}$, controlling for fluctuations in general response biases; S1 Text). Thus, while sharing the 1/f power law characteristic, fluctuations in stimulus- and history-congruence were shifted against each other by approximately half a cycle and showed a squared coherence of $6.49 \pm 2.07 \times 10^{-3}$% (Fig 2E and 2F; we report the average phase and coherence for frequencies below 0.1 $1/N_{trials}$; see Methods for details).

In sum, our analyses indicate that perceptual decisions result from a competition of external sensory signals with internal predictions provided by perceptual history. We show that the impact of these external and internal sources of information is not stable over time but fluctuates systematically, emitting overlapping autocorrelation curves and antiphase 1/f profiles.

These links between stimulus- and history-congruence suggest that the fluctuations in the impact of external and internal information are generated by a unifying mechanism that causes perception to alternate between 2 opposing *modes* [18] (Fig 1D): During *external mode*, perception is more strongly driven by the available external stimulus information. Conversely, during *internal mode*, participants rely more heavily on internal predictions that are implicitly provided by preceding perceptual experiences. The fluctuations in the degree of bias toward external versus internal information created by such *bimodal inference* may thus provide a novel explanation for ongoing fluctuations in the sensitivity to external information [4,5,18].

## 2.3 Internal and external modes of processing facilitate response behavior and enhance confidence in human perceptual decision-making

The above results point to systematic fluctuations in the *decision variable* [44] that determines perceptual choices, causing enhanced sensitivity to external stimulus information during external mode and increased biases toward preceding choices during internal mode. As such, fluctuations in mode should influence downstream aspects of behavior and cognition that operate on the perceptual decision variable [44]. To test this hypothesis with respect to motor behavior and metacognition, we asked how bimodal inference relates to response times (RTs) and confidence reports.

With respect to RTs, we observed faster responses for stimulus-congruent as opposed to stimulus-incongruent choices ($\beta = -0.14 \pm 1.6 \times 10^{-3}$, T($1.99 \times 10^{6}$) = −85.84, $p < 2.2 \times 10^{-308}$; Fig 2G). Intriguingly, while controlling for the effect of stimulus-congruence, we found that history-congruent (as opposed to history-incongruent) choices were also characterized by faster responses ($\beta = -9.56 \times 10^{-3} \pm 1.37 \times 10^{-3}$, T($1.98 \times 10^{6}$) = −6.97, $p = 3.15 \times 10^{-12}$; Fig 2G).

When analyzing the speed of response against the mode of sensory processing (Fig 2H), we found that RTs were shorter during externally oriented perception ($\beta_1 = -11.07 \pm 0.55$, T($1.98 \times 10^{6}$) = −20.14, $p = 3.17 \times 10^{-90}$). Crucially, as indicated by a quadratic relationship between the mode of sensory processing and RTs ($\beta_2 = -19.86 \pm 0.52$, T($1.98 \times 10^{6}$) = −38.43, $p = 5 \times 10^{-323}$), participants became faster at indicating their perceptual decision when biases toward both internal and external mode grew stronger.

In analogy to the speed of response, confidence was higher for stimulus-congruent as opposed to stimulus-incongruent choices ($\beta = 0.04 \pm 1.18 \times 10^{-3}$, T($2.06 \times 10^{6}$) = 36.85, $p = 3.25 \times 10^{-297}$; Fig 2I). Yet, while controlling for the effect of stimulus-congruence, we found that history-congruence also increased confidence ($\beta = 0.48 \pm 1.38 \times 10^{-3}$, T($2.06 \times 10^{6}$) = 351.54, $p < 2.2 \times 10^{-308}$; Fig 2I).

When depicted against the mode of sensory processing (Fig 2J), subjective confidence was enhanced when perception was more externally oriented ($\beta_1 = 92.63 \pm 1$, T($2.06 \times 10^{6}$) = 92.89,

$p < 2.2 \times 10^{-308}$). Importantly, however, participants were more confident in their perceptual decision for stronger biases toward both internal and external mode ($\beta_2 = 39.3 \pm 0.94$, T $(2.06 \times 10^6) = 41.95$, $p < 2.2 \times 10^{-308}$). In analogy to RTs, subjective confidence thus showed a quadratic relationship to the mode of sensory processing (Fig 2J).

Consequently, our findings predict that human participants lack full metacognitive insight into how strongly external signals and internal predictions contribute to perceptual decision-making. Stronger biases toward perceptual history thus lead to 2 seemingly contradictory effects, more frequent errors (S1C Fig) and increasing subjective confidence (Fig 2I and 2J). This observation generates an intriguing prediction regarding the association of between-mode fluctuations and perceptual metacognition: Metacognitive efficiency should be lower in individuals who spend more time in internal mode, since their confidence reports are less predictive of whether the corresponding perceptual decision is correct. We computed each participant's M-ratio [45] (meta-$d'/d' = 0.85 \pm 0.02$) to probe this hypothesis independently of interindividual differences in perceptual performance. Indeed, we found that biases toward internal information (as defined by the average probability of history-congruence) were stronger in participants with lower metacognitive efficiency ($\beta = -2.98 \times 10^{-3} \pm 9.82 \times 10^{-4}$, T $(4.14 \times 10^3) = -3.03$, $p = 2.43 \times 10^{-3}$).

In sum, the above results indicate that reporting behavior and metacognition do not map linearly onto the mode of sensory processing. Rather, they suggest that slow fluctuations in the respective impact of external and internal information are most likely to affect perception at an early level of sensory analysis [46,47]. Such low-level processing may thus integrate perceptual history with external inputs into a decision variable [44] that influences not only perceptual choices but also the speed and confidence at which they are made.

In what follows, we probe alternative explanations for between-mode fluctuations, test for the existence of modes in mice, and propose a predictive processing model that explains fluctuations in mode by ongoing shifts in the precision afforded to external sensory information relative to internal predictions driven by perceptual history.

## 2.4 Fluctuations between internal and external mode cannot be reduced to general response biases or random choices

The core assumption of bimodal inference—that ongoing changes in the sensitivity to external information are driven by internal predictions induced via perceptual history—needs to be contrasted against 2 alternative hypotheses: When making errors, observers may not engage with the task and respond stereotypically, i.e., exhibit stronger general biases toward one of the 2 potential outcomes or simply choose randomly.

Logistic regression confirmed that perceptual history made a significant contribution to perception ($\beta = 0.11 \pm 5.79 \times 10^{-3}$, z = 18.53, $p = 1.1 \times 10^{-76}$) over and above the ongoing stream of external sensory information ($\beta = 2.2 \pm 5.87 \times 10^{-3}$, z = 375.11, $p < 2.2 \times 10^{-308}$) and general response biases toward one of the two possible outcomes ($\beta = 15.19 \pm 0.08$, z = 184.98, $p < 2.2 \times 10^{-308}$).

When eliminating perceptual history as a predictor of individual choices, Akaike information criterion (AIC; [48]) increased by $\delta_{AIC} = 1.64 \times 10^3$ (see S4 Fig for parameter- and model-level inference at the level of individual observers). Likewise, when eliminating slow fluctuations in history-congruence as a predictor of slow fluctuations in stimulus-congruence across trials, we observed an increase in AIC by $\delta_{AIC} = 7.06 \times 10^3$. These results provided model-level evidence against the null hypotheses that fluctuations in stimulus-congruence are driven exclusively by choice randomness or general response bias (see S1 Text and S5 Fig for an in-depth assessment of general response bias).

To confirm that changes in the sensitivity to external information are indicative of internal mode processing, we estimated full and history-dependent psychometric curves during internal, external, and across modes [21]. If, as we hypothesized, internal mode processing reflects an enhanced impact of perceptual history, one would expect a history-dependent increase in biases and lapses as well as a history-independent increase in threshold. Conversely, if internal mode processing were driven by random choices, one would expect a history-independent increase in lapses and threshold and no change in bias. In line with our prediction, we found that internal mode processing was associated with a history-dependent increase in bias and lapse as well as a history-independent increase in threshold (S1 Text and S6 Fig). This confirmed that internal mode processing is indeed driven by an enhanced impact of perceptual history.

In line with this, the quadratic relationship between mode and confidence (Fig 2J) suggested that biases toward internal information do not reflect a postperceptual strategy of repeating preceding choices when the subjective confidence in the perceptual decision is low. Moreover, while responses became faster with longer exposure to the experiments of the Confidence database, the frequency of history-congruent choices increased over time, speaking against the proposition that participants stereotypically repeat preceding choices when not yet familiar with the experimental task (S1 Text).

Taken together, our results argue against recurring intervals of low task engagement, which may be signaled by stereotypical or random responses, as an alternative explanation for the phenomenon that we identify as bimodal inference.

## 2.5 Mice fluctuate between external and internal modes of sensory processing

In a prominent functional explanation for serial dependencies [22–28,32,33,46], perceptual history is cast as an internal prediction that leverages the temporal autocorrelation of natural environments for efficient decision-making [30,31,34,35,41]. Since this autocorrelation is one of the most basic features of our sensory world, fluctuating biases toward preceding perceptual choices should not be a uniquely human phenomenon.

To test whether externally and internally oriented modes of processing exist beyond the human mind, we analyzed data on perceptual decision-making in mice that were extracted from the International Brain Laboratory (IBL) dataset [21]. We restricted our analyses to the *basic* task [21], in which mice responded to gratings of varying contrast that appeared either in the left or right hemifield with equal probability. We excluded sessions in which mice did not respond correctly to stimuli presented at a contrast above 50% in more than 80% of trials (see Methods for details), which yielded a final sample of $N$ = 165 adequately trained mice that went through 1.46 million trials.

We found perception to be stimulus-congruent in 81.37% ± 0.3% of trials (Fig 3A, upper panel). In line with humans, mice were biased toward perceptual history in 54.03% ± 0.17% of trials (T(164) = 23.65, $p$ = 9.98×10$^{-55}$; Figs 3A and S1D). Since the *basic* task of the IBL dataset presented stimuli at random in either the left or the right hemifield, we expected stronger biases toward perceptual history to decrease perceptual performance. Indeed, history-congruent choices were more frequent when perception was stimulus-incongruent (61.59% ± 0.07%) as opposed to stimulus-congruent (51.81% ± 0.02%, T(164) = 31.37, $p$ = 3.36×10$^{-71}$; T(164) = 31.37, $p$ = 3.36×10$^{-71}$; Fig 3A, lower panel), confirming that perceptual history was a source of bias [24,28,30,31,43] as opposed to a feature of the experimental paradigm.

At the group level, we found significant autocorrelations in both stimulus-congruence (42 consecutive trials) and history-congruence (8 consecutive trials; Fig 3B), while controlling for

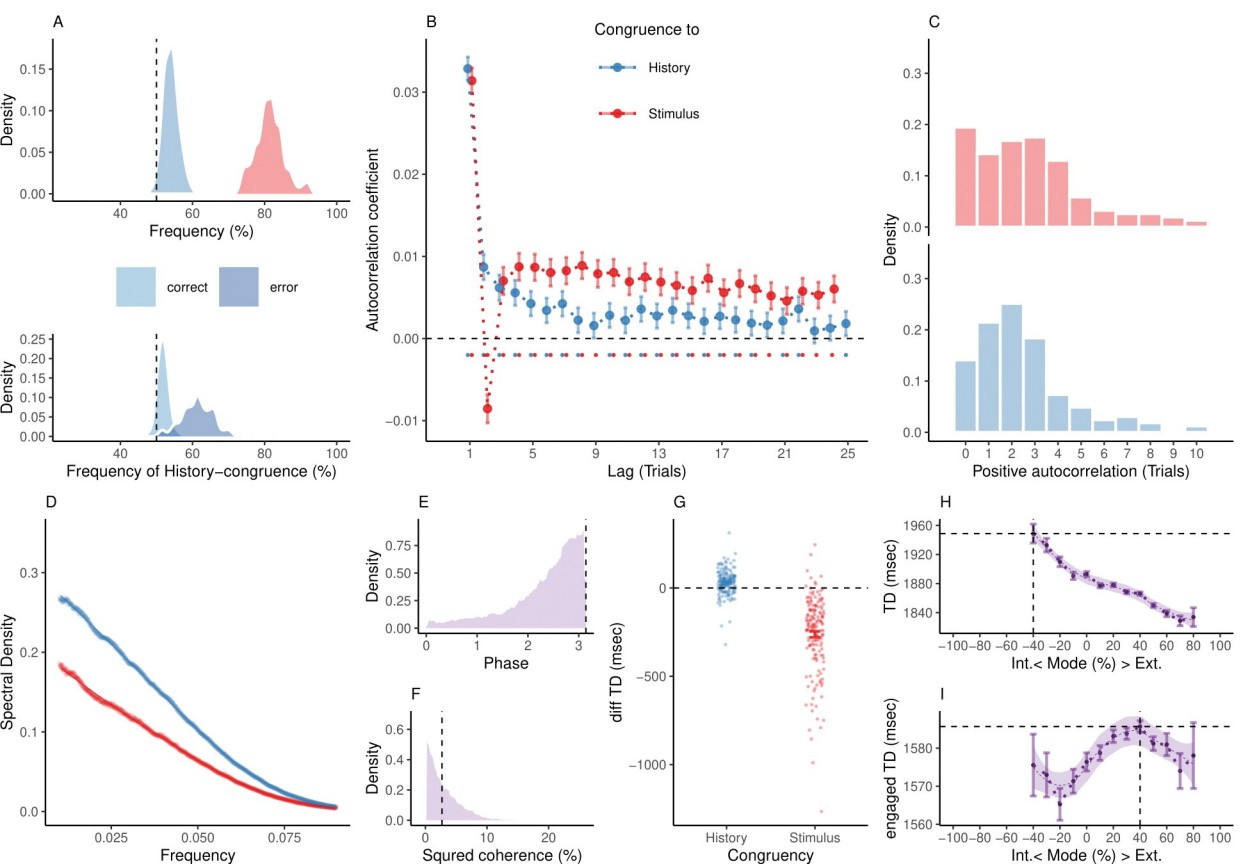

**Fig 3. Internal and external modes in mouse perceptual decision-making.** (**A**) In mice, 81.37% ± 0.3% of trials were stimulus-congruent (in red) and 54.03% ± 0.17% of trials were history-congruent (in blue; upper panel). History-congruent perceptual choices were not a consequence of the experimental design, but a source of error, as they were more frequent on stimulus-incongruent trials (lower panel). (**B**) Relative to randomly permuted data, we found highly significant autocorrelations of stimulus-congruence and history-congruence (dots indicate intercepts $\neq 0$ in trial-wise linear mixed effects modeling at $p < 0.05$). Please note that the negative autocorrelation of stimulus-congruence at trial 2 was a consequence of the experimental design (S2 Fig). As in humans, autocorrelation coefficients were best fit by an exponential function (adjusted $R^2$ for stimulus-congruence: 0.44; history-congruence: 0.52) as compared to a linear function (adjusted $R^2$ for stimulus-congruence: $3.16 \times 10^{-3}$; history-congruence: 0.26), decaying at a rate of $\gamma = -6.2 \times 10^{-4} \pm 5.93 \times 10^{-4}$ (T($3.55 \times 10^4$) = −1.05, $p = 0.3$) for stimulus-congruence and at a rate of $\gamma = -6.7 \times 10^{-3} \pm 5.94 \times 10^{-4}$ (T($3.69 \times 10^4$) = −11.27, $p = 2.07 \times 10^{-29}$) for history-congruence. (**C**) For stimulus-congruence (upper panel), the lag of positive autocorrelation was longer in comparison to humans (4.59 ± 0.06 on average). For history-congruence (lower panel), the lag of positive autocorrelation was slightly shorter relative to humans (2.58 ± 0.01 on average, peaking at trial t+2 after the index trial). (**D**) In mice, the dynamic probabilities of stimulus- and history-congruence (sliding windows of ±5 trials) fluctuated as a scale-invariant process with a 1/f power law. (**E**) The distribution of phase shift between fluctuations in stimulus- and history-congruence peaked at half a cycle ($\pi$ denoted by dotted line). (**F**) The average squared coherence between fluctuations in stimulus- and history-congruence (dotted line) amounted to 3.45 ± 0.01%. (**G**) We observed shorter trial durations (TDs) for stimulus-congruence (as opposed to stimulus-incongruence, $\beta = -1.12 \pm 8.53 \times 10^{-3}$, T($1.34 \times 10^6$) = −131.78, $p < 2.2 \times 10^{-308}$), but longer TDs for history-congruence ($\beta = 0.06 \pm 6.76 \times 10^{-3}$, T($1.34 \times 10^6$) = 8.52, $p = 1.58 \times 10^{-17}$). (**H**) TDs decreased monotonically for stronger biases toward external mode ($\beta_1 = -4.16 \times 10^4 \pm 1.29 \times 10^3$, T($1.35 \times 10^6$) = −32.31, $p = 6.03 \times 10^{-229}$). The horizontal and vertical dotted lines indicate maximum TD and the associated mode, respectively. (**I**) For TDs that differed from the median TD by no more than $1.5 \times$ MAD (median absolute distance; [49]), mice exhibited a quadratic component in the relationship between the mode of sensory processing and TDs ($\beta_2 = -1.97 \times 10^3 \pm 843.74$, T($1.19 \times 10^6$) = −2.34, $p = 0.02$). This explorative post hoc analysis focuses on trials at which mice engage more swiftly with the experimental task. The horizontal and vertical dotted lines indicate maximum TD and the associated mode, respectively.

the respective autocorrelation of task difficulty and external stimulation (S2 Fig). In contrast to humans, mice showed a negative autocorrelation coefficient of stimulus-congruence at trial 2, which was due to a feature of the experimental design: Errors at a contrast above 50% were followed by a high-contrast stimulus at the same location. Thus, stimulus-incongruent choices on easy trials were more likely to be followed by stimulus-congruent perceptual choices that were facilitated by high-contrast visual stimuli [21].

At the level of individual mice, autocorrelation coefficients were elevated above randomly permuted data within a lag of $4.59 \pm 0.06$ trials for stimulus-congruence and $2.58 \pm 0.01$ trials for history-congruence (Fig 3C). We corroborated these autocorrelations in logistic regression models that successfully predicted the stimulus-/history-congruence of perception at the index trial $t = 0$ from the stimulus-/history-congruence at the 33 preceding trials for stimulus-congruence and 8 preceding trials for history-congruence (S3 Fig). In analogy to humans, mice showed antiphase 1/f fluctuations in the sensitivity to internal and external information (Fig 3D–3F).

The above results confirm that fluctuations between internally and externally biased modes generalize to perceptual decision-making in mice. Following our hypothesis that bimodal inference operates at the level of perception, we predicted that between-mode fluctuations modulate a decision variable [44] that determines not only perceptual choices but also downstream aspects of mouse behavior [44]. We therefore asked how external and internal modes relate to the trial duration (TD, a coarse measure of RT in mice that spans the interval from stimulus onset to feedback; [21]). Stimulus-congruent (as opposed to stimulus-incongruent) choices were associated with shorter TDs ($\delta = -262.48 \pm 17.1$, T(164) = -15.35, $p = 1.55 \times 10^{-33}$), while history-congruent choices were characterized by longer TDs ($\delta = 30.47 \pm 5.57$, T(164) = 5.47, $p = 1.66 \times 10^{-7}$; Fig 3G).

Across the full spectrum of the available data, TDs showed a linear relationship with the mode of sensory processing, with shorter TDs during external mode ($\beta_1 = -4.16 \times 10^4 \pm 1.29 \times 10^3$, T($1.35 \times 10^6$) = $-32.31$, $p = 6.03 \times 10^{-229}$, Fig 3H). However, an explorative post hoc analysis limited to TDs that differed from the median TD by no more than $1.5 \times$ MAD (median absolute distance; [49]) indicated that, when mice engaged with the task more swiftly, TDs did indeed show a quadratic relationship with the mode of sensory processing ($\beta_2 = -1.97 \times 10^3 \pm 843.74$, T($1.19 \times 10^6$) = $-2.34$, $p = 0.02$, Fig 3I).

As in humans, it is important to ensure that ongoing changes in the sensitivity to external information are driven by perceptual history and cannot be reduced to general choice biases or random behavior. Logistic regression confirmed a significant effect of perceptual history on perceptual choices ($\beta = 0.51 \pm 4.49 \times 10^{-3}$, z = 112.84, $p < 2.2 \times 10^{-308}$), while controlling for external sensory information ($\beta = 2.96 \pm 4.58 \times 10^{-3}$, z = 646.1, $p < 2.2 \times 10^{-308}$) and general response biases toward one of the 2 outcomes ($\beta = -1.78 \pm 0.02$, z = $-80.64$, $p < 2.2 \times 10^{-308}$). When eliminating perceptual history as a predictor of individual choices, AIC increased by $\delta_{AIC} = 1.48 \times 10^4$, arguing against the notion that choice randomness and general response bias are the only determinants of perceptual performance in mice (see S4 Fig for parameter- and model-level inference in individual subjects).

In mice, fluctuations in the strength of history-congruent biases had a significant effect on stimulus-congruence ($\beta_1 = -0.12 \pm 7.17 \times 10^{-4}$, T($1.34 \times 10^6$) = $-168.39$, $p < 2.2 \times 10^{-308}$) beyond the effect of ongoing changes in general response biases ($\beta_2 = -0.03 \pm 6.94 \times 10^{-4}$, T($1.34 \times 10^6$) = $-48.14$, $p < 2.2 \times 10^{-308}$). Eliminating the dynamic fluctuations in history-congruence as a predictor of fluctuations in stimulus-congruence resulted in an increase in AIC by $\delta_{AIC} = 2.8 \times 10^4$ (see S1 Text and S5 Fig for an in-depth assessment of general response bias).

When fitting full and history-conditioned psychometric curves to the IBL data [21], we observed that internal mode processing was associated with a history-dependent increase in bias and lapse as well as a history-independent increase in threshold (S1 Text and S7 Fig). Over time, the frequency of history-congruent choices increased alongside stimulus-congruence and speed of response as mice were exposed to the experiment, arguing against the proposition that biases toward perceptual history reflected an unspecific response strategy in mice who were not sufficiently trained on the IBL task (S1 Text and S8 Fig).

In sum, these analyses confirmed that the observed fluctuations in sensitivity to external sensory information are driven by dynamic changes in the impact of perceptual history and cannot be reduced to general response bias and random choice behavior.

## 2.6 Fluctuations in mode result from coordinated changes in the impact of external and internal information on perception

The empirical data presented above indicate that, for both humans and mice, perception fluctuates between external and modes, i.e., multitrial epochs that are characterized by enhanced sensitivity toward either external sensory information or internal predictions generated by perceptual history. Since natural environments typically show high temporal redundancy [34], previous experiences are often good predictors of new stimuli [30,31,35,41]. Serial dependencies may therefore induce autocorrelations in perception by serving as internal predictions (or *memory* processes; [9,13]) that actively integrate noisy sensory information over time [50].

Previous work has shown that such internal predictions can be built by dynamically updating the estimated probability of being in a particular perceptual state from the sequence of preceding experiences [35,46,51]. The integration of sequential inputs may lead to accumulating effects of perceptual history that progressively override incoming sensory information, enabling internal mode processing [19]. However, since such a process would lead to internal biases that may eventually become impossible to overcome [52], changes in mode may require ongoing wave-like fluctuations [9,13] in the perceptual impact of external and internal information that occur *irrespective* of the sequence of previous experiences and temporarily decouple the decision variable from implicit internal representations of the environment [19].

Following Bayes' theorem, binary perceptual decisions depend on the log posterior ratio $L$ of the 2 alternative states of the environment that participants learn about via noisy sensory information [51]. We computed the posterior by combining the sensory evidence available at time point $t$ (i.e., the log likelihood ratio $LLR$) with the prior probability $\psi$, weighted by the respective precision terms $\omega_{LLR}$ and $\omega_{\psi}$:

$$L_t = LLR_t * \omega_{LLR} + \psi_t(L_{t-1}, H) * \omega_{\psi} \tag{1}$$

We derived the prior probability $\psi$ at time point $t$ from the posterior probability of perceptual outcomes at time point $L_{t-1}$. Since a switch between the 2 states can occur at any time, the effect of perceptual history varies according to both the sequence of preceding experiences and the estimated stability of the external environment (i.e., the *hazard rate H*; [51]):

$$\psi_t(L_{t-1}, H) = L_{t-1} + log\left(\frac{1-H}{H} + exp(-L_{t-1})\right) - log\left(\frac{1-H}{H} + exp(L_{t-1})\right) \tag{2}$$

The $LLR$ was computed from inputs $s_t$ by applying a sigmoid function defined by parameter $\alpha$ that controls the sensitivity of perception to the available sensory information (see Methods for details on $s_t$ in humans and mice):

$$u_t = \frac{1}{1 + exp(-\alpha * s_t)} \tag{3}$$

$$LLR_t = log\left(\frac{u_t}{1 - u_t}\right) \tag{4}$$

To allow for bimodal inference, i.e., alternating periods of internally and externally biased modes of perceptual processing that occur irrespective of the sequence of preceding

experiences, we assumed that likelihood and prior vary in their influence on the perceptual decision according to fluctuations governed by $\omega_{LLR}$ and $\omega_\psi$. These antiphase sine functions (defined by amplitudes $a_{LLR/\psi}$, frequency $f$, and phase $p$) determine the precision afforded to the likelihood and prior [53]. The implicit antiphase fluctuations are mandated by Bayes-optimal formulations in which inference depends only on the relative values of prior and likelihood precision (i.e., the Kalman gain; [54]). As such, $\omega_{LLR}$ and $\omega_\psi$ implement a hyperprior [55] in which the likelihood and prior precisions are shifted against each other at a dominant timescale defined by $f$:

$$\omega_{LLR} = a_{LLR}*sin(f*t + p) + 1 \tag{5}$$

$$\omega_\psi = a_\psi*sin(f*t + p + \pi) + 1 \tag{6}$$

Finally, a sigmoid transform of the posterior $L_t$ yields the probability of observing the perceptual decision $y_t$ at a temperature determined by $\zeta^{-1}$:

$$P(y_t = 1) = 1 - P(y_t = 0) = \frac{1}{1 + exp(-\zeta*L_t)} \tag{7}$$

We used a maximum likelihood procedure to fit the bimodal inference model (M1; Fig 1F) to the behavioral data from the Confidence database [20] and the IBL database [21], optimizing the parameters $\alpha$, $H$, $amp_{LLR}$, $amp_\psi$, $f$, $p$, and $\zeta$ (see Methods for details and S2 Table for a summary of the parameters of the bimodal inference model). We validated our model in 3 steps:

First, to show that bimodal inference does not emerge spontaneously in normative Bayesian models of evidence accumulation but requires the ad hoc addition of antiphase oscillations in prior and likelihood precision, we compared the bimodal inference model to 4 control models (M2 to M5; Fig 1G). In these models, we successively removed the antiphase oscillations (M2 to M4) and the integration of information across trials (M5) from the bimodal inference model and performed a model comparison based on AIC.

Model M2 ($AIC_2 = 9.76\times10^4$ in humans and $4.91\times10^4$ in mice) and Model M3 ($AIC_3 = 1.19\times10^5$ in humans and $5.95\times10^4$ in mice) incorporated only oscillations of either likelihood or prior precision. Model M4 ($AIC_4 = 1.69\times10^5$ in humans and $9.12\times10^4$ in mice) lacked any oscillations of likelihood and prior precision and corresponded to the normative model proposed by Glaze and colleagues [51]. In model M5 ($AIC_4 = 2.01\times10^5$ in humans and $1.13\times10^5$ in mice), we furthermore removed the integration of information across trials, such that perception depended only in incoming sensory information (Fig 1G).

The bimodal inference model achieved the lowest AIC across the full model space ($AIC_1 = 8.16\times10^4$ in humans and $4.24\times10^4$ in mice) and was clearly superior to the normative Bayesian model of evidence accumulation ($\delta_{AIC} = -8.79\times10^4$ in humans and $-4.87\times10^4$ in mice; S9 Fig).

As a second validation of the bimodal inference model, we tested whether the posterior model predicted within-training and out-of-training variables. The bimodal inference model characterizes each subject by a sensitivity parameter $\alpha$ (humans: $\alpha = 0.5 \pm 1.12\times10^{-4}$; mice: $\alpha = 1.06 \pm 2.88\times10^{-3}$) that captures how strongly perception is driven by the available sensory information, and a hazard rate parameter $H$ (humans: $H = 0.45 \pm 4.8\times10^{-5}$; mice: $H = 0.46 \pm 2.97\times10^{-4}$) that controls how heavily perception is biased by perceptual history. The parameter $f$ captures the dominant timescale at which likelihood (amplitude humans: $a_{LLR} = 0.5 \pm 2.02\times10^{-4}$; mice: $a_{LLR} = 0.39 \pm 1.08\times10^{-3}$) and prior precision (amplitude humans: $a_\psi = 1.44 \pm 5.27\times10^{-4}$; mice: $a_\psi = 1.71 \pm 7.15\times10^{-3}$) fluctuated and was estimated at $0.11 \pm 1.68\times10^{-5}$ $1/N_{trials}$ and $0.11 \pm 1.63\times10^{-4}$ $1/N_{trials}$ in mice.

As a sanity check for model fit, we tested whether the frequency of stimulus- and history-congruent trials in the Confidence database [20] and IBL database [21] correlated with the estimated parameters $\alpha$ and $H$, respectively. As expected, the estimated sensitivity toward stimulus information $\alpha$ was positively correlated with the frequency of stimulus-congruent perceptual choices (humans: $\beta = 0.84 \pm 0.26$, T($4.31 \times 10^3$) = 32.87, $p = 1.3 \times 10^{-211}$; mice: $\beta = 1.93 \pm 0.12$, T($2.07 \times 10^3$) = 16.21, $p = 9.37 \times 10^{-56}$). Likewise, $H$ was negatively correlated with the frequency of history-congruent perceptual choices (humans: $\beta = -11.84 \pm 0.5$, T($4.29 \times 10^3$) = −23.5, $p = 5.16 \times 10^{-115}$; mice: $\beta = -6.18 \pm 0.66$, T($2.08 \times 10^3$) = −9.37, $p = 1.85 \times 10^{-20}$).

Our behavioral analyses reveal that humans and mice show significant effects of perceptual history that impaired performance in randomized psychophysical experiments [24,28,30,31,43] (Figs 2A and 3A). We therefore expected that humans and mice underestimated the true hazard rate $\hat{H}$ of the experimental environments (Confidence database [20]: $\hat{H}_{Humans} = 0.5 \pm 1.58 \times 10^{-5}$); IBL database [21]: $\hat{H}_{Mice} = 0.49 \pm 6.48 \times 10^{-5}$). Indeed, when fitting the bimodal inference model to the trial-wise perceptual choices, we found that the estimated (i.e., subjective) hazard rate $H$ was lower than $\hat{H}$ for both humans ($\beta = -6.87 \pm 0.94$, T(61.87) = −7.33, $p = 5.76 \times 10^{-10}$) and mice ($\beta = -2.91 \pm 0.34$, T(112.57) = −8.51, $p = 8.65 \times 10^{-14}$).

To further probe the validity of the bimodal inference model, we asked whether posterior model quantities could explain aspects of the behavioral data that the model was not fitted to. We predicted that the posterior decision variable $L_t$ not only encodes perceptual choices (i.e., the variable used for model estimation) but also predicts the speed of response and subjective confidence [30,44]. Indeed, the estimated trial-wise posterior decision certainty $|L_t|$ correlated negatively with RTs in humans ($\beta = -4.36 \times 10^{-3} \pm 4.64 \times 10^{-4}$, T($1.98 \times 10^6$) = −9.41, $p = 5.19 \times 10^{-21}$) and TDs mice ($\beta = -35.45 \pm 0.86$, T($1.28 \times 10^6$) = −41.13, $p < 2.2 \times 10^{-308}$). Likewise, subjective confidence reports were positively correlated with the estimated posterior decision certainty in humans ($\beta = 7.63 \times 10^{-3} \pm 8.32 \times 10^{-4}$, T($2.06 \times 10^6$) = 9.18, $p = 4.48 \times 10^{-20}$).

The dynamic accumulation of information inherent to our model entails that biases toward perceptual history are stronger when the posterior decision certainty at the preceding trial is high [30,31,51]. Due to the link between posterior decision certainty and confidence, confident perceptual choices should be more likely to induce history-congruent perception at the subsequent trial [30,31]. In line with our prediction, logistic regression indicated that history-congruence was predicted by the posterior decision certainty $|L_{t-1}|$ extracted from the model (humans: $\beta = 8.22 \times 10^{-3} \pm 1.94 \times 10^{-3}$, z = 4.25, $p = 2.17 \times 10^{-5}$; mice: $\beta = -3.72 \times 10^{-3} \pm 1.83 \times 10^{-3}$, $z = -2.03$, $p = 0.04$) and the subjective confidence reported by the participants (humans: $\beta = 0.04 \pm 1.62 \times 10^{-3}$, $z = 27.21$, $p = 4.56 \times 10^{-163}$) at the preceding trial.

As a third validation of the bimodal inference model, we used the posterior model parameters to simulate synthetic perceptual choices and repeated the behavioral analyses conducted for the empirical data. Simulations from the bimodal inference model closely replicated our empirical results: Simulated perceptual decisions resulted from a competition of perceptual history with incoming sensory signals (Fig 4A). Stimulus- and history-congruence were significantly autocorrelated (Fig 4B and 4C), fluctuating in antiphase as a scale-invariant process with a 1/f power law (Fig 4D–4F). Simulated posterior certainty [28,30,44] (i.e., the absolute of the log posterior ratio $|L_t|$) showed a quadratic relationship to the mode of sensory processing (Fig 4H), mirroring the relation of RTs and confidence reports to external and internal biases in perception (Figs 2G, 2H, 3G and 3H,). Crucially, the overlap between empirical and simulated data broke down when we removed the antiphase oscillations or the accumulation of evidence over time from the bimodal inference model (S10–S13 Figs).

In sum, computational modeling suggested that between-mode fluctuations are best explained by 2 interlinked processes (Fig 1E and 1F): (i) the dynamic accumulation of

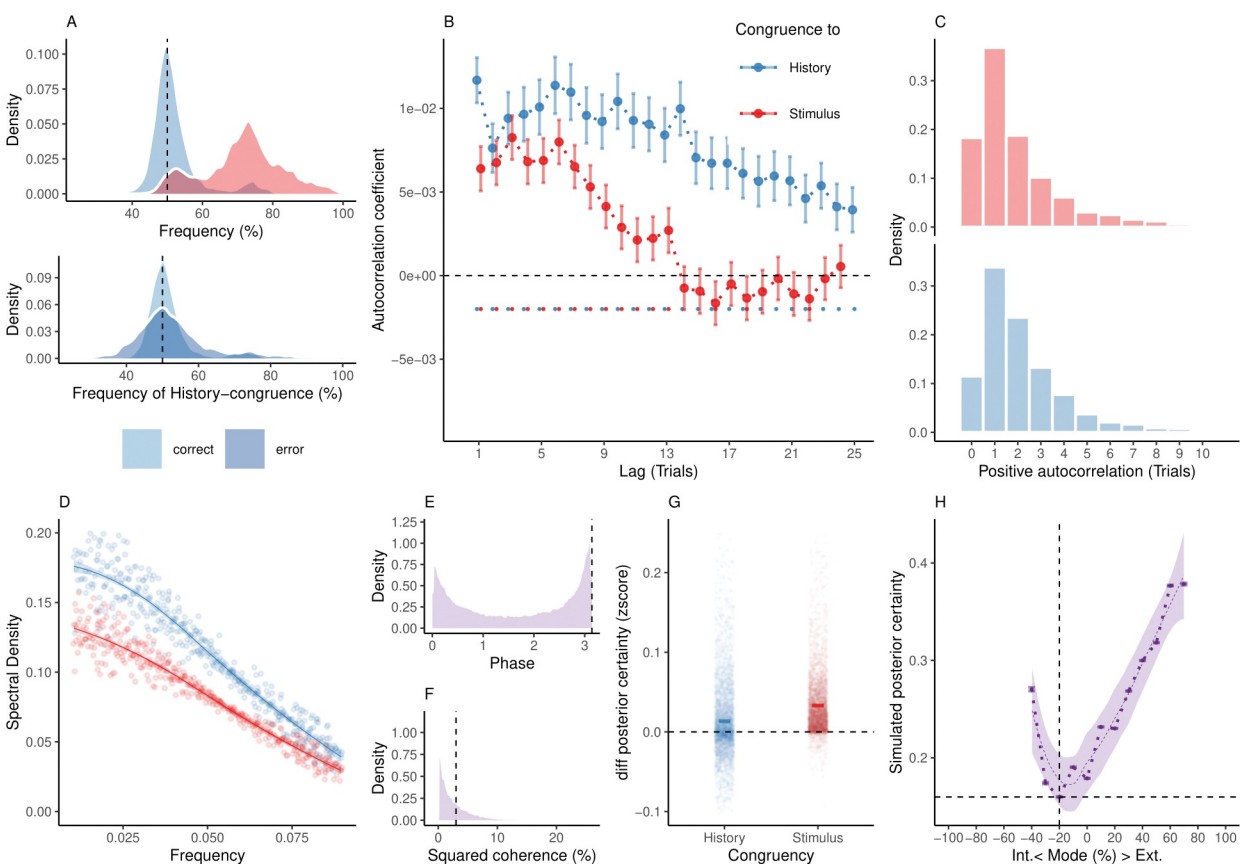

**Fig 4. Internal and external modes in simulated perceptual decision-making.** (**A**) Simulated perceptual choices were stimulus-congruent in 71.36% ± 0.17% (in red) and history-congruent in 51.99% ± 0.11% of trials (in blue; T(4.32×10³) = 17.42, p = 9.89×10⁻⁶⁶; upper panel). Due to the competition between stimulus- and history-congruence, history-congruent perceptual choices were more frequent when perception was stimulus-incongruent (i.e., on *error* trials; T(4.32×10³) = 11.19, p = 1.17×10⁻²⁸; lower panel) and thus impaired performance in the randomized psychophysical design simulated here. (**B**) At the simulated group level, we found significant autocorrelations in both stimulus-congruence (13 consecutive trials) and history-congruence (30 consecutive trials). (**C**) On the level of individual simulated participants, autocorrelation coefficients exceeded the autocorrelation coefficients of randomly permuted data within a lag of 2.46 ± 1.17×10⁻³ trials for stimulus-congruence and 4.24 ± 1.85×10⁻³ trials for history-congruence. (**D**) The smoothed probabilities of stimulus- and history-congruence (sliding windows of ±5 trials) fluctuated as a scale-invariant process with a 1/f power law, i.e., at power densities that were inversely proportional to the frequency (power ∼ 1/fᵝ; stimulus-congruence: β = −0.81 ± 1.18×10⁻³, T(1.92×10⁵) = −687.58, p < 2.2×10⁻³⁰⁸; history-congruence: β = −0.83 ± 1.27×10⁻³, T(1.92×10⁵) = −652.11, p < 2.2×10⁻³⁰⁸). (**E**) The distribution of phase shift between fluctuations in simulated stimulus- and history-congruence peaked at half a cycle (π denoted by dotted line). The dynamic probabilities of simulated stimulus- and history-congruence were therefore were strongly anticorrelated (β = −0.03 ± 8.22×10⁻⁴, T(2.12×10⁶) = −40.52, p < 2.2×10⁻³⁰⁸). (**F**) The average squared coherence between fluctuations in simulated stimulus- and history-congruence (black dotted line) amounted to 6.49 ± 2.07×10⁻³%. (**G**) Simulated confidence was enhanced for stimulus-congruence (β = 0.03 ± 1.71×10⁻⁴, T(2.03×10⁶) = 178.39, p < 2.2×10⁻³⁰⁸) and history-congruence (β = 0.01 ± 1.5×10⁻⁴, T(2.03×10⁶) = 74.18, p < 2.2×10⁻³⁰⁸). (**H**) In analogy to humans, the simulated data showed a quadratic relationship between the mode of perceptual processing and posterior certainty, which increased for stronger external and internal biases (β₂ = 31.03 ± 0.15, T(2.04×10⁶) = 205.95, p < 2.2×10⁻³⁰⁸). The horizontal and vertical dotted lines indicate minimum posterior certainty and the associated mode, respectively.

information across successive trials mandated by normative Bayesian models of evidence accumulation and (ii) ongoing antiphase oscillations in the impact of external and internal information.

## 3. Discussion

This work investigates the behavioral and computational characteristics of ongoing fluctuations in perceptual decision-making using 2 large-scale datasets in humans [20] and mice [21]. We found that humans and mice cycle through recurring intervals of reduced sensitivity to

external sensory information, during which they rely more strongly on perceptual history, i.e., an internal prediction that is provided by the sequence of preceding choices. Computational modeling indicated that these slow periodicities are governed by 2 interlinked factors: (i) the dynamic integration of sensory inputs over time and (ii) antiphase oscillations in the strength at which perception is driven by internal versus external sources of information. These cross-species results suggest that ongoing fluctuations in perceptual decision-making arise not merely as a noise-related epiphenomenon of limited processing capacity but result from a structured and adaptive mechanism that fluctuates between internally and externally oriented modes of sensory analysis.

### 3.1 Bimodal inference represents a pervasive aspect of perceptual decision-making in humans and mice

A growing body of literature has highlighted that perception is modulated by preceding choices [22–28,30,32,33]. Our work provides converging cross-species evidence supporting the notion that such serial dependencies are a pervasive and general phenomenon of perceptual decision-making (Figs 2 and 3). While introducing errors in randomized psychophysical designs [24,28,30,31,43] (Figs 2A and 3A), we found that perceptual history facilitates postperceptual processes such as speed of response [42] (Figs 2G and 3G) and subjective confidence in humans (Fig 2I).

At the level of individual traits, increased biases toward preceding choices were associated with reduced sensitivity to external information (S1 Fig) and lower metacognitive efficiency. When investigating how serial dependencies evolve over time, we observed dynamic changes in the strength of perceptual history (Figs 2 and 3B) that created wavering biases toward internally and externally biased modes of sensory processing. Between-mode fluctuations may thus provide a new explanation for ongoing changes in perceptual performance [6–11].

In computational terms, serial dependencies may leverage the temporal autocorrelation of natural environments [31,46] to increase the efficiency of decision-making [35,43]. Such temporal smoothing [46] of sensory inputs may be achieved by updating dynamic predictions about the world based on the sequence of noisy perceptual experiences [22,31], using algorithms based on sequential Bayes [25,42,51] such as Kalman [35] or Hierarchical Gaussian filtering [54]. At the level of neural mechanisms, the integration of internal with external information may be realized by combining feedback from higher levels in the cortical hierarchy with incoming sensory signals that are fed forward from lower levels [56].

Yet, relying too strongly on serial dependencies may come at a cost: When accumulating over time, internal predictions may eventually override external information, leading to circular and false inferences about the state of the environment [57]. Akin to the wake–sleep algorithm in machine learning [58], bimodal inference may help to determine whether errors result from external input or from internally stored predictions: During internal mode, sensory processing is more strongly constrained by predictive processes that auto-encode the agent's environment. Conversely, during external mode, the network is driven predominantly by sensory inputs [18]. Between-mode fluctuations may generate an unambiguous error signal that aligns internal predictions with the current state of the environment in iterative test-update cycles [58]. On a broader scale, between-mode fluctuations may regulate the balance between feedforward versus feedback contributions to perception and thereby play an adaptive role in metacognition and reality monitoring [59].

We hypothesized that observers have certain hyperpriors that are apt for accommodating fluctuations in the predictability of their environment, i.e., people believe that their world is inherently volatile. To be Bayes optimal, it is therefore necessary to periodically reevaluate

posterior beliefs about the parameters that define an internal generative model of the external sensory environment. One way to do this is to periodically suspend the precision of prior beliefs and increase the precision afforded to sensory evidence, thus updating Bayesian beliefs about model parameters.

The empirical evidence above suggests that the timescale of this periodic scheduling of evidence accumulation may be scale-invariant. This means that there may exist a timescale of periodic fluctuations in precision over every window or length of perceptual decision-making. Bimodal inference predicts perceptual decisions under a generative model (based upon a hazard function to model serial dependencies between subsequent trials) with periodic fluctuations in the precision of sensory evidence relative to prior beliefs at a particular timescale. Remarkably, a systematic model comparison based on AIC indicated that a model with fluctuating precisions has much greater evidence, relative to a model in the absence of fluctuating precisions. This ad hoc addition of oscillations to a normative Bayesian model of evidence accumulation [51] allowed us to quantify the dominant timescale of periodic fluctuations mode at approximately 0.11 $1/N_{trials}$ in humans and mice that is appropriate for these kinds of paradigms.

## 3.2 Bimodal inference versus normative Bayesian evidence accumulation

Could bimodal inference emerge spontaneously in normative models of perceptual decision-making? In predictive processing, the relative precision of prior and likelihood determines their integration into the posterior that determines the content of perception. At the level of individual trials, the perceptual impact of internal predictions generated from perceptual history (prior precision) and external sensory information (likelihood precision) are thus necessarily anticorrelated. The same holds for mechanistic models of drift diffusion, which understand choice history biases as driven by changes in the starting point [51] or the drift rate of evidence accumulation [32]. Under the former formulation, perceptual history is bound to have a stronger influence on perception when less weight is given to incoming sensory evidence, assuming that the last choice is represented as a starting point bias. The effects of choice history in normative Bayesian and mechanistic drift diffusion models can be mapped onto one another via the Bayesian formulation of drift diffusion [60], where the inverse of likelihood precision determines the amount of noise in the accumulation of new evidence, and prior precision determines the absolute shift in its starting point [60].

While it is thus clear that the impact of perceptual history and sensory evidence are anticorrelated *at each individual trial*, we here introduce antiphase oscillations as an ad hoc modification to model slow fluctuations in prior and likelihood precision that evolve *over many consecutive trials* and are not mandated by normative Bayesian or mechanistic drift diffusion models. The bimodal inference model provides a reasonable explanation of the linked autocorrelations in stimulus- and history-congruence, as evidenced by formal model comparison, successful prediction of RTs and confidence as out-of-training variables, and a qualitative reproduction of our empirical data from posterior model parameter as evidence against over- or underfitting.

Of note, similar non-stationarities have been observed in descriptive models that assume continuous [61] or discrete [12] changes in the latent states that modulate perceptual decision-making at slow timescales. A recent computational study [12] has used a Hidden Markov model to investigate perceptual decision-making in the IBL database [21]. In analogy to our findings, the authors observed that mice switch between temporally extended *strategies* that last for more than 100 trials: During *engaged* states, perception was highly sensitive to external sensory information. During *disengaged* states, in turn, choice behavior was prone to errors

due to enhanced biases toward one of the 2 perceptual outcomes [12]. Despite the conceptual differences to our approach (discrete states in a Hidden Markov model that correspond to switches between distinct decision-making strategies [12] versus gradual changes in mode that emerge from sequential Bayesian inference and ongoing oscillations in the impact of external relative to internal information), it is tempting to speculate that engaged/disengaged states and between-mode fluctuations might tap into the same underlying phenomenon.

## 3.3 Task engagement and residual motor activation as alternative explanations for bimodal inference

As a functional explanation for bimodal inference, we propose that perception temporarily disengages from internal predictions to form stable inferences about the statistical properties of the sensory environment. Between-mode fluctuations may thus elude circular inferences that occur when both the causes and the encoding of sensory stimuli are volatile [19,57]. By the same token, we suggest that fluctuations in mode occur at the level of perceptual processing [26,30,46,47] and are not a passive phenomenon that is primarily driven by factors situated up- or downstream of sensory analysis.

How does attention relate to phenomenon of between-mode fluctuations? According to predictive processing, attention corresponds to the precision afforded to the probability distributions that underlie perceptual inference [53]. From this perspective, fluctuations between external and internal mode can be understood as ongoing shifts in the attention afforded to either external sensory information (regulated via likelihood precision) or internal predictions (regulated via prior precision). When the precision of either likelihood or prior increases, posterior precision increases, which leads to faster RTs and higher confidence. Therefore, when defined from the perspective of predictive processing as the precision afforded to likelihood and prior [53], fluctuations in attention may provide a plausible explanation for the quadratic relationship of mode to RTs and confidence (Figs 2H–2J, 3I, and 4I).

Outside of the predictive processing field, attention is often understood in the context of task engagement [62], which varies according to the availability of cognitive resources that are modulated by factors such as tonic arousal, familiarity with the task, or fatigue [62]. Our results suggest that internal mode processing cannot be completely reduced to intervals of low task engagement: In addition to shorter RTs and elevated confidence, choices during internal mode were not random or globally biased but driven by perceptual history (S1 Text). Moreover, our computational model identified the dominant timescale of between-mode fluctuations at 0.11 $1/N_{trials}$, which may be compatible with fluctuations in arousal [63] but is faster than to be expected for the development of task familiarity or fatigue.

However, in interpreting the impact of between-mode fluctuations on perceptual accuracy, speed of response, and confidence, it is important to consider that global modulators such as tonic arousal are known to have nonlinear effects on task performance [64]: In perceptual tasks, performance seems so be highest during midlevel arousal, whereas low- and high-level arousal lead to reduced accuracy and slower responses [64]. This contrasts with the effects of bimodal inference, where accuracy increases linearly as one moves from internal to external mode, and responses become faster at both ends of the mode spectrum.

Of note, high phasic arousal has been shown to suppress biases in decision-making in humans and mice across domains [65–67], including biases toward perceptual history [28] that we implicate in internal mode processing. While the increase in response speed and history congruence over time (S1 Text) may argue against insufficient training as an alternative explanation for internal mode processing, it may also be indicative of waning arousal. The multiple mechanistic mappings to RTs and confidence warrant more direct measures of

arousal (such as pupil size [28,63–65,67,68], motor behavior [63,68], or neural data [69]) to better delineate bimodal inference from fluctuations in global modulators of task performance.

Residual activation of the motor system may provide another contribution to serial biases in perceptual choices [70]. Such motor-driven priming may lead to errors in randomized psychophysical designs, resembling the phenomenon that we identify as internally biased processing [71]. Moreover, residual activation of the motor system may lead to faster responses and thus constitutes an alternative explanation for the quadratic relationship of mode with RTs [70]. The observation of elevated confidence for stronger biases toward internal mode speaks against the proposition that residual activation of the motor system is the primary driver of serial choice biases, since strong motor-driven priming should lead to frequent lapses that are typically associated with reduced confidence [72]. Likewise, perceptual history effects have repeatedly been replicated in experiments with counterbalanced stimulus–response mappings [30].

No-response paradigms, in which perceptual decisions are inferred from eye movements alone, could help to better differentiate perceptual from motor-related effects. Likewise, video tracking of response behavior and neural recordings from motor- and premotor cortex, which has recently been released for the IBL database [21], may provide further insight into the relation of motor behavior to the perceptual phenomenon of between-mode fluctuations.

## 3.4 Limitations and open questions

Our results suggest bimodal inference as a pervasive aspect of perceptual decision-making in humans and mice. However, a number of limitations and open questions have to be considered:

First, this work sought to understand whether fluctuations between internal and external mode, which we initially observed in an experiment on bistable perception in humans [19], represent a general phenomenon that occurs across a diverse set of perceptual decision-making tasks. Our analysis of the Confidence database [20] therefore collapsed across all available experiments on binary perceptual decision-making. Individual experiments differed with respect to the stimuli, the manipulation of difficulty, the timing of trials, and the way responses were collected but were highly comparable with respect to the central variables of stimulus- and history-congruence (S1 Fig).

The variability across experiments, which we considered as random effects in all statistical analyses, enabled us to assess whether bimodal inference represents a general phenomenon in perceptual decision-making but limited the precision at which we were able to investigate the relation of mode to behavioral variables such as timing, task difficulty, RT, or confidence. This issue is partially resolved by our analyses of the IBL database, which replicated our findings in an experiment that was highly standardized with respect to timing, task difficulty, and behavioral readout [21]. It will be an important task for future research to validate our results on bimodal inference in a standardized dataset of comparable volume in humans, which is, to our knowledge, not yet available.

Second, our results point to an attraction of perception toward preceding choices. Previous work has shown that perceptual decision-making is concurrently affected by both attractive and repulsive serial biases that operate on distinct timescales and serve complementary functions for sensory processing [27,73,74]: Short-term attraction may serve the decoding of noisy sensory inputs and increase the stability of perception, whereas long-term repulsion may enable efficient encoding and sensitivity to change [27]. In the data analyzed here, history biases tended to be repetitive (Figs 2A, 3A, S6, and S7), and only 2 of the 66 experiments of the

Confidence database [20] showed significant alternating biases (S1 Fig). However, as we show in S14 Fig, fluctuations in both alternating and repeating history biases generate overlapping autocorrelation curves. Our analysis of between-mode fluctuations is therefore not tied exclusively to repeating biases but accommodates alternating biases as well, such that both may lead to internally biased processing and reduced sensitivity to external sensory information. Future work could apply our approach to paradigms that boost alternating as opposed to repeating biases, as this would help to better understand how repetition and alternation are linked in terms of their computational function and neural implementation [27].

A third open question concerns the computational underpinnings of bimodal inference. The addition of slow antiphase oscillations to the integration of prior and likelihood represents an ad hoc modification of a normative Bayesian model of evidence accumulation [51]. While the bimodal inference model is supported by formal model comparison, the successful prediction of out-of-training variables, and the qualitative reproduction of our empirical data in simulations from posterior model parameters, it is an important task for future research to test (i) whether between-mode fluctuations can emerge spontaneously in hierarchical models of Bayesian inference, (ii) whether modes are continuous [19] or discrete [12], and (iii) whether bimodal inference can be causally manipulated by experimental variables. We speculate that between-mode fluctuations may separate the perceptual contribution of internal predictions and external sensory data in time, creating unambiguous learning signals that benefit inference about the precision of prior and likelihood, respectively. This proposition should be tested empirically by relating the phenomenon of bimodal inference to performance in, e.g., reversal learning, probabilistic reasoning, or metacognition.

A final important avenue for further research on bimodal inference is to elucidate its neurobiological underpinnings. Since between-mode fluctuations were found in humans and mice, future studies can apply noninvasive and invasive neuroimaging and electrophysiology to better understand the neural mechanisms that generate ongoing changes in mode in terms of their neuroanatomy, neurochemistry, and neurocircuitry.

Establishing the neural correlates of externally and internally biased modes will enable exiting opportunities to investigate their role for adaptive perception and decision-making: Causal interventions via pharmacological challenges, optogenetic manipulations, or (non)invasive brain stimulation will help to understand whether between-mode fluctuations are implicated in resolving credit-assignment problems [18,75] or in calibrating metacognition and reality monitoring [59]. Answers to these questions may provide new insights into the pathophysiology of hallucinations and delusions, which have been characterized by an imbalance in the impact of external versus internal information [56,76,77] and are typically associated with metacognitive failures and a departure from consensual reality [77].

## 4. Methods

### 4.1 Resource availability

**4.1.1 Lead contact.**   Further information and requests for resources should be directed to and will be fulfilled by the lead contact, Veith Weilnhammer (veith.weilnhammer@gmail.com).

**4.1.2 Materials availability.**   This study did not generate new unique reagents.

### 4.2 Experimental model and subject details

**4.2.1 Confidence database.**   We downloaded the human data from the Confidence database [20] on October 21, 2020, limiting our analyses to the category *perception*. Within this category, we selected studies in which participants made binary perceptual decisions between 2

alternatives. We excluded 2 experiments in which the average perceptual accuracy fell below 50%. After excluding these experiments, our sample consisted of 21.05 million trials obtained from 4,317 human participants and 66 individual experiments (S1 Table). Out of the 66 included experiments, 62 investigated visual, 1 auditory, 2 proprioceptive, and 1 multimodal perception. A total of 59 experiments were based on discrimination and 6 on detection, with one investigating both.

**4.2.2 IBL database.** We downloaded the data from the IBL database [21] on April 28, 2021. We limited our analyses to the *basic task*, during which mice responded to gratings that appeared with equal probability in the left or right hemifield. Within each mouse, we excluded sessions in which perceptual accuracy was below 80% for stimuli presented at a contrast ≥50%. After exclusion, our sample consisted of 1.46 million trials trials obtained from $N = 165$ mice.

## 4.3 Method details

**4.3.1 Variables of interest.   Primary variables of interest:** We extracted trial-wise data on the presented stimulus and the associated perceptual decision. Stimulus-congruent choices were defined by perceptual decisions that matched the presented stimuli. History-congruent choices were defined by perceptual choices that matched the perceptual choice at the immediately preceding trial. The dynamic probabilities of stimulus- and history-congruence were computed in sliding windows of ±5 trials.

The *mode* of sensory processing was derived by subtracting the dynamic probability of history-congruence from the dynamic probability of stimulus-congruence, such that positive values indicate externally oriented processing, whereas negative values indicate internally oriented processing. When visualizing the relation of the mode of sensory processing to confidence, RTs, or TD (see below), we binned the mode variable in 10% intervals. We excluded bins that contained less than 0.5% of the total number of available data-points.

**Secondary variables of interest**: From the Confidence database [20], we furthermore extracted trial-wise confidence reports and RTs. Out of the 58 experiments that provide information on RTs, 46 cued the response by the onset of a response screen or an additional response cue, whereas 14 allowed participants to response at any time after stimulus onset. If RTs were available for both the perceptual decision and the confidence report, we only extracted the RT associated with the perceptual decision. To enable comparability between studies, we normalized RTs and confidence reports within individual studies using the *scale* R function. If not available for a particular study, RTs were treated as missing variables. From the IBL database [21], we extracted TDs as defined by interval between stimulus onset and feedback, which represents a coarse measure of RT [21].

**Exclusion criteria for individual data points:** For nonnormalized data (TDs from the IBL database [21]; d-prime, meta-dprime, and M-ratio from the Confidence database [20] and simulated confidence reports), we excluded data points that differed from the median by more than $3 \times$ MAD [49]. For normalized data (RTs and confidence reports from the Confidence database [20]), we excluded data points that differed from the mean by more than $3 \times$ SD (standard deviation).

**4.3.2 Control variables.   ** Next to the sequence of presented stimuli, we assessed the autocorrelation of task difficulty as an alternative explanation for any autocorrelation in stimulus- and history-congruence. In the Confidence database [20], 21 of the 66 included experiments used fixed difficulty levels, whereas 45 manipulated difficulty levels within participants. Difficulty was manipulated via noise masks, contrast, luminance, presentation time, stimulus probability for gabors, dot coherence for random dot kinematograms, difference in elements and

set size for comparisons of numerosity, difference in clicks for auditory discrimination, temporal distance for meta-contrast masking, and amount of self-motion for proprioception. We treated task difficulty as a missing variable for the experiments that fixed it at the participant level, as this precluded the computation of autocorrelation curves. In analogy to RTs and confidence, difficulty levels were normalized within individual studies. For the IBL database [21], task difficulty was defined by the contrast of the presented grating.

**4.3.3 Autocorrelations.** For each participant, trial-wise autocorrelation coefficients were estimated using the R function *acf* with a maximum lag defined by the number of trials available per subject. Autocorrelation coefficients are displayed against the lag (in numbers of trials, ranging from 1 to 20) relative to the index trial (t = 0; Figs 2B, 2C, 3B, 3C, 4B, and 4C). To account for spurious autocorrelations that occur due to imbalances in the analyzed variables, we estimated autocorrelations for randomly permuted data (100 iterations). For group-level autocorrelations, we computed the differences between the true autocorrelation coefficients and the mean autocorrelation observed for randomly permuted data and averaged across participants.

At a given trial, group-level autocorrelation coefficients were considered significant when linear mixed effects modeling indicated that the difference between real and permuted autocorrelation coefficients was above 0 at an alpha level of 0.05%. To test whether the autocorrelation of stimulus- and history-congruence remained significant when controlling for task difficulty and the sequence of presented stimuli, we added the respective autocorrelation as an additional factor to the linear mixed effects model that computed the group-level statistics (see also Mixed effects modeling).

To assess autocorrelations at the level of individual participants, we counted the number of subsequent trials (starting at the first trial after the index trial) for which less than 50% of the permuted autocorrelation coefficients exceeded the true autocorrelation coefficient. For example, a count of 0 indicates that the true autocorrelation coefficients exceeded *less than 50%* of the autocorrelation coefficients computed for randomly permuted data at the first trial following the index trial. A count of 5 indicates that, for the first 5 trials following the index trial, the true autocorrelation coefficients exceeded *more than 50%* of the respective autocorrelation coefficients for the randomly permuted data; at the sixth trial following the index trial, however, *less than 50%* of the autocorrelation coefficients exceeded the respective permuted autocorrelation coefficients.

**4.3.4 Spectral analysis.** We used the R function *spectrum* to compute the spectral densities for the dynamic probabilities of stimulus- and history-congruence as well as the phase (i.e., frequency-specific shift between the 2 time-series ranging from 0 to $2^*\pi$) and squared coherence (frequency-specific variable that denotes the degree to which the shift between the 2 time-series in constant, ranging from 0% to 100%). Periodograms were smoothed using modified Daniell smoothers at a width of 50.

Since the dynamic probabilities of history- and stimulus-congruence were computed using a sliding windows of ±5 trials (i.e., intervals containing a total of 11 trials), we report the spectral density, coherence, and phase for frequencies below 0.1 $1/N_{trials}$. Spectral densities have 1 value per subject and frequency (data shown in Figs 2D and 3D). To assess the relation between stimulus- and history-congruence in this frequency range, we report average phase and average squared coherence for all frequencies below 0.1 $1/N_{trials}$ (i.e., 1 value per subject; data shown in Figs 2E, 2F, 3E, and 3F).

Since the data extracted from the Confidence database [20] consist of a large set of individual studies that differ with respect to intertrial intervals, we defined the variable *frequency* in the dimension of cycles per trial $1/N_{trials}$ rather than cycles per second (Hz). For consistency, we chose $1/N_{trials}$ as the unit of frequency for the IBL database [21] as well.

## 4.4 Quantification and statistical procedures

All aggregate data are reported and displayed with errorbars as mean ± standard error of the mean.

**4.4.1 Mixed effects modeling.** Unless indicated otherwise, we performed group-level inference using the R packages *lmer* and *afex* for linear mixed effects modeling and *glmer* with a binomial link function for logistic regression. We compared models based on AIC. To account for variability between the studies available from the Confidence database [20], mixed modeling was conducted using random intercepts defined for each study. To account for variability across experimental session within the IBL database [21], mixed modeling was conducted using random intercepts defined for each individual session. When multiple within-participant data points were analyzed, we estimated random intercepts for each participant that were *nested* within the respective study of the Confidence database [20]. By analogy, for the IBL database [21], we estimated random intercepts for each session that were nested within the respective mouse. We report $\beta$ values referring to the estimates provided by mixed effects modeling, followed by the respective T statistic (linear models) or z statistic (logistic models).

The effects of stimulus- and history-congruence on RTs and confidence reports (Figs 2–4, subpanels G-I) were assessed in linear mixed effects models that tested for main effects of both stimulus- and history-congruence as well as the between-factor interaction. Thus, the significance of any effect of history-congruence on RTs and confidence reports was assessed while controlling for the respective effect of stimulus-congruence (and vice versa).

**4.4.2 Psychometric function.** We obtained psychometric curves by fitting the following error function to the behavioral data:

$$y_p = \gamma + (1 - \gamma - \delta) * \left( erf\left(\frac{s_w + \mu}{t}\right) + 1 \right)/2 \tag{8}$$

We used the Broyden–Fletcher–Goldfarb–Shanno algorithm in maximum likelihood estimation [78] to predict individual choices $y$ (outcome A: $y = 0$; outcome B: $y = 1$) from the choice probability $y_p$. In humans, we computed $s_w$ by multiplying the inputs $s$ (stimulus A: 0; outcome B: 1) with the task difficulty $D_b$ (binarized across 7 levels):

$$s_w = (s - 0.5) * D_b \tag{9}$$

In mice, $s_w$ was defined by the respective stimulus contrast in the 2 hemifields:

$$s_w = Contrast_{Right} - Contrast_{Left} \tag{10}$$

Parameters of the psychometric error function were fitted using the R package *optimx* [78]. The psychometric error function was defined via the parameters $\gamma$ (lower lapse; lower bound = 0, upper bound = 0.5), $\delta$ (upper lapse; lower bound = 0, upper bound = 0.5), $\mu$ (bias; lower bound humans = −5; upper bound humans = 5, lower bound mice = −0.5, upper bound mice = 0.5), and threshold $t$ (lower bound humans = 0.5, upper bound humans = 25; lower bound mice = 0.01, upper bound mice = 1.5).

**4.4.3 Computational modeling. Model definition**: Our modeling analysis is an extension of a model proposed by Glaze and colleagues [51], who defined a normative account of evidence accumulation for decision-making. In this model, trial-wise choices are explained by applying Bayes theorem to infer moment-by-moment changes in the state of environment from trial-wise noisy observations across trials.

Following Glaze and colleagues [51], we applied Bayes rule to compute the posterior evidence for the 2 alternative choices (i.e., the log posterior ratio $L$) from the sensory evidence available at time point $t$ (i.e., the log likelihood ratio $LLR$) with the prior probability $\psi$,

weighted by the respective precision terms $\omega_{LLR}$ and $\omega_\psi$:

$$L_t = LLR_t * \omega_{LLR} + \psi_t(L_{t-1}, H) * \omega_\psi \tag{11}$$

In the trial-wise design studied here, a transition between the 2 states of the environment (i.e., the sources generating the noisy observations available to the participant) can occur at any time. Despite the random nature of the psychophysical paradigms studied here [20,21], humans and mice showed significant biases toward preceding choices [79] (Figs 2A and 3A). We thus assumed that the prior probability of the 2 possible outcomes depends on the posterior choice probability at the preceding trial and the hazard rate $H$ assumed by the participant. Following Glaze and colleagues [51], the prior $\psi$ is thus computed as follows:

$$\psi_t(L_{t-1}, H) = L_{t-1} + log\left(\frac{1-H}{H} + exp(-L_{t-1})\right) - log\left(\frac{1-H}{H} + exp(L_{t-1})\right) \tag{12}$$

In this model, humans, mice, and simulated agents make perceptual choices based on noisy observations $u$. The are computed by applying a sensitivity parameter $\alpha$ to the content of external sensory information $s$. For humans, we defined the input $s$ by the 2 alternative states of the environment (stimulus A: $s = 0$; stimulus B: $s = 1$), which generated the observations $u$ through a sigmoid function that applied a sensitivity parameter $\alpha$:

$$u_t = \frac{1}{1 + exp(-\alpha*(s_t - 0.5))} \tag{13}$$

In mice, the inputs $s$ were defined by the respective stimulus contrast in the 2 hemifields:

$$s_t = Contrast_{Right} - Contrast_{Left} \tag{14}$$

As in humans, we derived the input $u$ by applying a sigmoid function with a sensitivity parameter $\alpha$ to input $s$:

$$u_t = \frac{1}{1 + exp(-\alpha*s_t)} \tag{15}$$

For humans, mice, and in simulations, the log likelihood ratio $LLR$ was computed from $u$ as follows:

$$LLR_t = log\left(\frac{u_t}{1 - u_t}\right) \tag{16}$$

To allow for long-range autocorrelation in stimulus- and history-congruence (Figs 2B and 3B), our modeling approach differed from Glaze and colleagues [51] in that it allowed for systematic fluctuation in the impact of sensory information (i.e., $LLR$) and the prior probability of choices $\psi$ on the posterior probability $L$. This was achieved by multiplying the log likelihood ratio and the log prior ratio with coherent antiphase fluctuations according to $\omega_{LLR} = a_{LLR}*sin(f*t + phase) + 1$ and $\omega_\psi = a_\psi*sin(f*t + phase + \pi) + 1$.

**Model fitting**: In model fitting, we predicted the trial-wise choices $y_t$ (option A: 0; option B: 1) from inputs $s$. To this end, we minimized the log loss between $y_t$ and the choice probability $y_{pt}$ in the unit interval. $y_{pt}$ was derived from $L_t$ using a sigmoid function defined by the inverse

decision temperature $\zeta$:

$$y_{pt} = \frac{1}{1 + exp(-\zeta * L_t)} \tag{17}$$

This allowed us to infer the free parameters $H$ (lower bound = 0, upper bound = 1; human posterior = $0.45 \pm 4.8 \times 10^{-5}$; mouse posterior = $0.46 \pm 2.97 \times 10^{4}$), $\alpha$ (lower bound = 0, upper bound = 5; human posterior = $0.5 \pm 1.12 \times 10^{-4}$; mouse posterior = $1.06 \pm 2.88 \times 10^{-3}$), $a_{\psi}$ (lower bound = 0, upper bound = 10; human posterior = $1.44 \pm 5.27 \times 10^{-4}$; mouse posterior = $1.71 \pm 7.15 \times 10^{-3}$), $amp_{LLR}$ (lower bound = 0, upper bound = 10; human posterior = $0.5 \pm 2.02 \times 10^{-4}$; mouse posterior = $0.39 \pm 1.08 \times 10^{-3}$), frequency $f$ (lower bound = 1/40, upper bound = 1/5; human posterior = $0.11 \pm 1.68 \times 10^{-5}$; mouse posterior = $0.11 \pm 1.63 \times 10^{-4}$), $p$ (lower bound = 0, upper bound = $2*\pi$; human posterior = $2.72 \pm 4.41 \times 10^{-4}$; mouse posterior = $2.83 \pm 3.95 \times 10^{-3}$), and inverse decision temperature $\zeta$ (lower bound = 1, upper bound = 10; human posterior = $4.63 \pm 1.95 \times 10^{-4}$; mouse posterior = $4.82 \pm 3.03 \times 10^{-3}$) using maximum likelihood estimation with the Broyden–Fletcher–Goldfarb–Shanno algorithm as implemented in the R function *optimx* [78] (see S2 Table for a description of our model parameters).

We validated the bimodal inference model in 3 steps: a formal model comparison to reduced models based on AIC (Figs 1F, 1G, and S9), the prediction of within-training (stimulus- and history-congruence) as well as out-of-training variables (RT and confidence), and a qualitative reproduction of the empirical data from model simulations based on estimated parameters (Fig 4).

**Model comparison.** We assessed the following model space based on AIC:

- The full *bimodal inference model* (M1; Fig 1F) incorporates the influence of sensory information according to the parameter $\alpha$ (likelihood), the integration of evidence across trials according to the parameter $H$ (prior), antiphase oscillations in between likelihood and prior precision according to $\omega_{LLR}$ and $\omega_{\psi}$ with parameters $a_{LLR}$ (amplitude likelihood fluctuation), $a_{\psi}$ (amplitude prior fluctuation), $f$ (frequency), and $p$ (phase).

- The *likelihood-oscillation-only model* (M2; Fig 1G) incorporates the influence of sensory information according to parameter $\alpha$ (likelihood), the integration of evidence across trials according to parameter $H$ (prior), oscillations in likelihood precision according to $\omega_{LLR}$ with parameters $a_{LLR}$ (amplitude likelihood fluctuation), $f$ (frequency), and $p$ (phase).

- The *prior-oscillation-only model* (M3; Fig 1G) incorporates the influence of sensory information according to parameter $\alpha$ (likelihood), the integration of evidence across trials according to parameter $H$ (prior), oscillations in the prior precision according to $\omega_{\psi}$ with parameters $a_{\psi}$ (amplitude prior fluctuation), $f$ (frequency), and $p$ (phase). Please note that all models M1 to M3 lead to shifts in the relative precision of likelihood and prior.

- The *normative-evidence-accumulation model* (M4; Fig 1G) incorporates the influence of sensory information according to parameter $\alpha$ (likelihood) and the integration of evidence across trials according to parameter $H$ (prior). There are no additional oscillations. Model M4 thus corresponds to the model proposed by Glaze and colleagues and captures normative evidence accumulation in unpredictable environments using a Bayesian update scheme [51]. The comparison against M4 tests the null hypothesis that fluctuations in mode emerge from a normative Bayesian model without the ad hoc addition of oscillations as in models M1 to M3.

- The *no-evidence-accumulation model* (M5; Fig 1G) incorporates the influence of sensory information according to parameter $\alpha$ (likelihood). The model lacks integration of evidence across trials (flat prior) and oscillations. The comparison against M5 tests the null hypothesis that observers do not use prior information derived from serial dependency in perception.

**Prediction of within-training and out-of-training variables.** To validate our model, we correlated individual posterior parameter estimates with the respective conventional variables. As a sanity check, we tested (i) whether the estimated hazard rate $H$ correlated negatively with the frequency of history-congruent choices and (ii) whether the estimated sensitivity to sensory information $\alpha$ correlated positively with the frequency of stimulus-congruent choices. In addition, we tested whether the posterior decision certainty (i.e., the absolute of the log posterior ratio) correlated negatively with RTs and positively with confidence. This allowed us to assess whether our model could explain aspects of the data it was not fitted to (i.e., RTs and confidence).

**Simulations.** Finally, we used simulations (Figs 4 and S10–S13) to show that all model components, including the antiphase oscillations governed by $a_\psi$, $a_{LLR}$, $f$, and $p$, were necessary for our model to reproduce the characteristics of the empirical data. This enabled us to assess over- or underfitting in the bimodal inference model and all reduced models M2 to M5. We used the posterior model parameters observed for humans ($H$, $\alpha$, $a_\psi$, $a_{LLR}$, $f$, $p$, and $\zeta$) to define individual parameters for simulation in 4,317 simulated participants (i.e., equivalent to the number of human participants). For each participant, the number of simulated trials was drawn at random between 300 and 700. Inputs $s$ were drawn at random for each trial, such that the sequence of inputs to the simulation did not contain any systematic seriality. Noisy observations $u$ were generated by applying the posterior parameter $\alpha$ to inputs $s$, thus generating stimulus-congruent choices in $71.36 \pm 2.6 \times 10^{-3}$% of trials. Choices were simulated based on the trial-wise choice probabilities $y_p$ obtained from our model. Simulated data were analyzed in analogy to the human and mouse data. As a substitute of subjective confidence, we computed the absolute of the trial-wise log posterior ratio $|L|$ (i.e., the posterior decision certainty).

## Supporting information

**S1 Text. Choice history, general response bias, psychometric functions, and task familiarity.** In this supplemental file, we show that internal mode processing is driven by choice history as opposed to stimulus history, that fluctuations between internal and external mode modulate perceptual performance beyond the effect of general response biases, that internal mode is characterized by lower thresholds as well as by history-dependent changes in biases and lapses, and that internal mode processing can not be reduced to insufficient task familiarity.
(PDF)

**S1 Fig. Stimulus- and history-congruence.** (**A**) Stimulus-congruent choices in humans amounted to 73.46% ± 0.15% of trials and were highly consistent across the experiments selected from the Confidence database. (**B**) History-congruent choices in humans amounted to 52.7% ± 0.12% of trials. In analogy to stimulus-congruence, the prevalence of history-congruence was highly consistent across the experiments selected from the Confidence database. A percentage of 48.48% of experiments showed significant ($p < 0.05$) biases toward preceding choices, whereas 2 of the 66 of the included experiments showed significant repelling biases. (**C**) In humans, we found an enhanced impact of perceptual history in participants who were less sensitive to external sensory information ($T(4.3 \times 10^3) = -14.27$, $p = 3.78 \times 10^{-45}$), suggesting

that perception results from the competition of external with internal information. (**D**) In analogy to humans, mice that were less sensitive to external sensory information showed stronger biases toward perceptual history (T(163) = −7.52, $p$ = 3.44×10$^{-12}$, Pearson correlation).
(TIFF)

**S2 Fig. Controlling for task difficulty and external stimulation.** In this study, we found highly significant autocorrelations of stimulus- and history-congruence in humans as well as in mice, while controlling for task difficulty and the sequence of external stimulation. Here, we confirm that the autocorrelations of stimulus- and history-congruence were not a trivial consequence of the experimental design or the addition of task difficulty and external stimulation as control variables in the computation of group-level autocorrelations. (**A**) In humans, task difficulty (in green) showed a significant autocorrelation starting at the fifth trial (upper panel, dots at the bottom indicate intercepts $\neq$ 0 in trial-wise linear mixed effects modeling at $p$ < 0.05). When controlling for task difficulty only, linear mixed effects modeling indicated a significant autocorrelation of stimulus-congruence (in red) for the first 3 consecutive trials (middle panel). Around 20% of trials within the displayed time window remained significantly autocorrelated. The autocorrelation of history-congruence (in blue) remained significant for the first 11 consecutive trials (64% significantly autocorrelated trials within the displayed time window). At the level of individual participants, the autocorrelation of task difficulty exceeded the respective autocorrelation of randomly permuted within a lag of 21.66 ± 8.37×10$^{-3}$ trials (lower panel). (**B**) In humans, the sequence of external stimulation (i.e., which of the 2 binary outcomes was supported by the presented stimuli; depicted in green) was negatively autocorrelated for 1 trial. When controlling for the autocorrelation of external stimulation only, stimulus-congruence remained significantly autocorrelated for 22 consecutive trials (88% of trials within the displayed time window; lower panel) and history-congruence remained significantly autocorrelated for 20 consecutive trials (84% of trials within the displayed time window). At the level of individual participants, the autocorrelation of external stimulation exceeded the respective autocorrelation of randomly permuted within a lag of 2.94 ± 4.4×10$^{-3}$ consecutive trials (lower panel). (**C**) In mice, task difficulty showed a significant autocorrelation for the first 25 consecutive trials (upper panel). When controlling only for task difficulty, linear mixed effects modeling indicated a significant autocorrelation of stimulus-congruence for the first 36 consecutive trials (middle panel). In total, 100% of trials within the displayed time window remained significantly autocorrelated. The autocorrelation of history-congruence remained significant for the first 8 consecutive trials, with 84% significantly autocorrelated trials within the displayed time window. At the level of individual mice, autocorrelation coefficients for difficulty were elevated above randomly permuted data within a lag of 15.13 ± 0.19 consecutive trials (lower panel). (**D**) In mice, the sequence of external stimulation (i.e., which of the 2 binary outcomes was supported by the presented stimuli) was negatively autocorrelated for 11 consecutive trials (upper panel). When controlling only for the autocorrelation of external stimulation, stimulus-congruence remained significantly autocorrelated for 86 consecutive trials (100% of trials within the displayed time window; middle) and history-congruence remained significantly autocorrelated for 8 consecutive trials (84% of trials within the displayed time window). At the level of individual mice, autocorrelation coefficients for external stimulation were elevated above randomly permuted data within a lag of 2.53 ± 9.8×10$^{-3}$ consecutive trials (lower panel).
(TIFF)

**S3 Fig. Reproducing group-level autocorrelations using logistic regression.** (**A**) As an alternative to group-level autocorrelation coefficients, we used trial-wise logistic regression to

quantify serial dependencies in stimulus- and history-congruence. This analysis predicted stimulus- and history-congruence at the index trial (trial $t = 0$, vertical line) based on stimulus- and history-congruence at the 100 preceding trials. Mirroring the shape of the group-level autocorrelations, trial-wise regression coefficients (depicted as mean ± SEM, dots mark trials with regression weights significantly greater than 0 at $p < 0.05$) increased toward the index trial $t = 0$ for the human data. (**B**) Following our results in human data, regression coefficients that predicted history-congruence at the index trial (trial t = 0, vertical line) increased exponentially for trials closer to the index trial in mice. In contrast to history-congruence, stimulus-congruence showed a negative regression weight (or autocorrelation coefficient; Fig 3B) at trial −2. This was due to the experimental design (see also the autocorrelations of difficulty and external stimulation in S2 Fig): When mice made errors at easy trials (contrast $\geq 50\%$), the upcoming stimulus was shown at the same spatial location and at high contrast. This increased the probability of stimulus-congruent perceptual choices after stimulus-incongruent perceptual choices at easy trials, thereby creating a negative regression weight (or autocorrelation coefficient) of stimulus-congruence at trial −2.
(TIFF)

**S4 Fig. History-congruence in logistic regression.** (**A**) To ensure that perceptual history played a significant role in perception despite the ongoing stream of external information, we tested whether human perceptual decision-making was better explained by the combination of external and internal information or, alternatively, by external information alone. To this end, we compared AIC between logistic regression models that predicted trial-wise perceptual responses either by both current external sensory information and the preceding percept or by external sensory information alone (values above 0 indicate a superiority of the full model). With high consistency across the experiments selected from the Confidence Database, this model comparison confirmed that perceptual history contributed significantly to perception (difference in AIC = 8.07 ± 0.53, $T(57.22) = 4.1$, $p = 1.31 \times 10^{-4}$). (**B**) Participant-wise regression coefficients amount to 0.18 ± 0.02 for the effect of perceptual history and 2.51 ± 0.03 for external sensory stimulation. (**C**) In mice, an AIC-based model comparison indicated that perception was better explained by logistic regression models that predicted trial-wise perceptual responses based on both current external sensory information and the preceding percept (difference in AIC = 88.62 ± 8.57, $T(164) = -10.34$, $p = 1.29 \times 10^{-19}$). (**D**) In mice, individual regression coefficients amounted to 0.42 ± 0.02 for the effect of perceptual history and 6.91 ± 0.21 for external sensory stimulation.
(TIFF)

**S5 Fig. Correcting for general response biases.** Here, we ask whether the autocorrelation of history-congruence (as shown in Figs 2–3C) may be driven by general response biases (i.e., a general propensity to choose one of the 2 possible outcomes more frequently than the alternative). To this end, we generated sequences of 100 perceptual choices with general response biases ranging from 60% to 90% for 1,000 simulated participants each. We then computed the autocorrelation of history-congruence for these simulated data. Crucially, we used the correction procedure that is applied to the autocorrelation curves shown in this manuscript: All reported autocorrelation coefficients are computed relative to the average autocorrelation coefficients obtained for 100 iterations of randomly permuted trial sequences. The above simulation show that this correction procedure removes any potential contribution of general response biases to the autocorrelation of history-congruence. This indicates that the autocorrelation of history-congruence (as shown in Figs 2–3C) is not driven by general response biases that were present in the empirical data at a level of 58.71% ± 0.22% in humans and 54.6% ±

0.3% in mice.
(TIFF)

**S6 Fig. Full and history-conditioned psychometric functions across modes in humans.** (**A**) Here, we show average psychometric functions for the full dataset (upper panel) and conditioned on perceptual history ($y_{t-1} = 1$ and $y_{t-1} = 0$; middle and lower panel) across modes (green line) and for internal mode (blue line) and external mode (red line) separately. (**B**) Across the full dataset, biases $\mu$ were distributed around 0 ($\beta_0 = 7.37\times10^{-3} \pm 0.09$, T(36.8) = 0.08, $p = 0.94$; upper panel), with larger absolute biases $|\mu|$ for internal as compared to external mode ($\beta_0 = -0.62 \pm 0.07$, T(45.62) = -8.38, $p = 8.59\times10^{-11}$; controlling for differences in lapses and thresholds). When conditioned on perceptual history, we observed negative biases for $y_{t-1}$ = 0 ($\beta_0 = 0.56 \pm 0.12$, T(43.39) = 4.6, $p = 3.64\times10^{-5}$; middle panel) and positive biases for $y_{t-1}$ = 1 ($\beta_0 = 0.56 \pm 0.12$, T(43.39) = 4.6, $p = 3.64\times10^{-5}$; lower panel). (**C**) Lapse rates were higher in internal mode as compared to external mode ($\beta_0 = -0.05 \pm 5.73\times10^{-3}$, T(47.03) = -9.11, $p = 5.94\times10^{-12}$; controlling for differences in biases and thresholds; see upper panel and sub-plot D). Importantly, the between-mode difference in lapses depended on perceptual history: We found no significant difference in lower lapses $\gamma$ for $y_{t-1}$ = 0 ($\beta_0 = 0.01 \pm 7.77\times10^{-3}$, T(33.1) = 1.61, $p = 0.12$; middle panel), but a significant difference for $y_{t-1}$ = 1 ($\beta_0 = -0.11 \pm 0.01$, T(40.11) = -9.59, $p = 6.14\times10^{-12}$; lower panel). (**D**) Conversely, higher lapses $\delta$ were significantly increased for $y_{t-1}$ = 0 ($\beta_0 = -0.1 \pm 9.58\times10^{-3}$, T(36.87) = -10.16, $p = 3.06\times10^{-12}$; middle panel), but not for $y_{t-1}$ = 1 ($\beta_0 = 0.01 \pm 7.74\times10^{-3}$, T(33.66) = 1.58, $p = 0.12$; lower panel). (**E**) The thresholds $t$ were larger in internal as compared to external mode ($\beta_0 = -1.77 \pm 0.25$, T(50.45) = -7.14, $p = 3.48\times10^{-9}$; controlling for differences in biases and lapses) and were not modulated by perceptual history ($\beta_0 = 0.04 \pm 0.06$, T(2.97×10³) = 0.73, $p = 0.47$).
(TIFF)

**S7 Fig. Full and history-conditioned psychometric functions across modes in mice.** (**A**) Here, we show average psychometric functions for the full IBL dataset (upper panel) and conditioned on perceptual history ($y_{t-1} = 1$ and $y_{t-1} = 0$; middle and lower panel) across modes (green line) and for internal mode (blue line) and external mode (red line) separately. (**B**) Across the full dataset, biases $\mu$ were distributed around 0 (T(164) = 0.39, $p = 0.69$; upper panel), with larger absolute biases $|\mu|$ for internal as compared to external mode ($\beta_0 = -0.18 \pm 0.03$, T = -6.38, $p = 1.77\times10^{-9}$; controlling for differences in lapses and thresholds). When conditioned on perceptual history, we observed negative biases for $y_{t-1}$ = 0 (T(164) = -1.99, $p = 0.05$; middle panel) and positive biases for $y_{t-1}$ = 1 (T(164) = 1.91, $p = 0.06$; lower panel). (**C**) Lapse rates were higher in internal as compared to external mode ($\beta_0 = -0.11 \pm 4.39\times10^{-3}$, T = -2.48, $p = 4.91\times10^{-57}$; controlling for differences in biases and thresholds; upper panel, see subplot D). For $y_{t-1}$ = 1, the difference between internal and external mode was more pronounced for lower lapses $\gamma$ (T(164) = -18.24, $p = 2.68\times10^{-41}$) as compared to higher lapses $\delta$ (see subplot D). In mice, lower lapses $\gamma$ were significantly elevated during internal mode irrespective of the preceding perceptual choice (middle panel: lower lapses $\gamma$ for $y_{t-1}$ = 0; T(164) = -2.5, $p = 0.01$, lower panel: lower lapses $\gamma$ for $y_{t-1}$ = 1; T(164) = -32.44, $p = 2.92\times10^{-73}$). (**D**) For $y_{t-1}$ = 0, the difference between internal and external mode was more pronounced for higher lapses $\delta$ (T(164) = 21.44, $p = 1.93\times10^{-49}$; see subplot C). Higher lapses were significantly elevated during internal mode irrespective of the preceding perceptual choice (middle panel: higher lapses $\delta$ for $y_{t-1}$ = 0; T(164) = -28.29, $p = 5.62\times10^{-65}$ lower panel: higher lapses $\delta$ for $y_{t-1}$ = 1; T(164) = -2.65, $p = 8.91\times10^{-3}$). (**E**) Thresholds $t$ were higher in internal as compared to external mode ($\beta_0 = -0.28 \pm 0.04$, T = -7.26, $p = 1.53\times10^{-11}$; controlling for differences in biases and lapses) and were not modulated by perceptual history (T(164)

= 0.94, $p = 0.35$).
(TIFF)

**S8 Fig. History-/stimulus-congruence and TDs during training of the basic task.** Here, we depict the progression of history- and stimulus-congruence (depicted in blue and red, respectively; left panel) as well as TDs (in green; right panel) across training sessions in mice that achieved proficiency (i.e., stimulus-congruence $\geq 80\%$) in the *basic* task of the IBL dataset. We found that both history-congruent perceptual choices ($\beta = 0.13 \pm 4.67 \times 10^{-3}$, $\mathrm{T}(8.4 \times 10^3) = 27.04$, $p = 1.96 \times 10^{-154}$) and stimulus-congruent perceptual choices ($\beta = 0.34 \pm 7.13 \times 10^{-3}$, $\mathrm{T}(8.51 \times 10^3) = 47.66$, $p < 2.2 \times 10^{-308}$) became more frequent with training. As in humans, mice showed shorter TDs with increased exposure to the task ($\beta = -22.14 \pm 17.06$, $\mathrm{T}(1.14 \times 10^3) = -1.3$, $p < 2.2 \times 10^{-308}$).
(TIFF)

**S9 Fig. Comparison of the bimodal inference model against reduced control models.** (**A**) Group-level AIC. The bimodal inference model (M1) achieved the lowest AIC across the full model space ($AIC_1 = 8.16 \times 10^4$ in humans and $4.24 \times 10^4$ in mice). Model M2 ($AIC_2 = 9.76 \times 10^4$ in humans and $4.91 \times 10^4$ in mice) and Model M3 ($AIC_3 = 1.19 \times 10^5$ in humans and $5.95 \times 10^4$ in mice) incorporated only oscillations of either likelihood or prior precision. Model M4 ($AIC_4 = 1.69 \times 10^5$ in humans and $9.12 \times 10^4$ in mice) lacked any oscillations of likelihood and prior precision and corresponded to the normative model proposed by Glaze and colleagues [51]. In model M5 ($AIC_4 = 2.01 \times 10^5$ in humans and $1.13 \times 10^5$ in mice), we furthermore removed the integration of information across trials, such that perception depended only in incoming sensory information. (**B**) Subject-level AIC. Here, we show the distribution of AIC values at the subject level. AIC for the bimodal inference model tended to be smaller than AIC for the comparator models (statistical comparison to the second-best model M2 in humans: $\beta = -1.71 \pm 0.19$, $\mathrm{T}(8.57 \times 10^3) = -8.85$, $p = 1.06 \times 10^{-18}$; mice: $\mathrm{T}(1.57 \times 10^3) = -3.08$, $p = 2.12 \times 10^{-3}$).
(TIFF)

**S10 Fig. Reduced control model M2: Only oscillation of the likelihood.** When simulating data for the *likelihood-oscillation-only model*, we removed the oscillation from the prior term by setting the amplitude $a_\psi$ to 0. Simulated data thus depended only on the participant-wise estimates for hazard rate $H$, amplitude $a_{LLR}$, frequency $f$, phase $p$, and inverse decision temperature $\zeta$. (**A**) Similar to the full model M1 (Figs 1F and 4), simulated perceptual choices were stimulus-congruent in $71.97\% \pm 0.17\%$ of trials (in red). History-congruent amounted to $50.76\% \pm 0.07\%$ of trials (in blue). As in the full model, the likelihood-oscillation-only model showed a significant bias toward perceptual history $\mathrm{T}(4.32 \times 10^3) = 10.29$, $p = 1.54 \times 10^{-24}$; upper panel). Similarly, history-congruent choices were more frequent at error trials ($\mathrm{T}(4.32 \times 10^3) = 9.71$, $p = 4.6 \times 10^{-22}$; lower panel). (**B**) In the likelihood-oscillation-only model, we observed that the autocorrelation coefficients for history-congruence were reduced below the autocorrelation coefficients of stimulus-congruence. This is an approximately 5-fold reduction relative to the empirical results observed in humans (Fig 2B), where the autocorrelation of history-congruence was above the autocorrelation of stimulus-congruence. Moreover, in the reduced model shown here, the number of consecutive trials that showed significant autocorrelation of history-congruence was reduced to 11. (**C**) In the likelihood-oscillation-only model, the number of consecutive trials at which true autocorrelation coefficients exceeded the autocorrelation coefficients for randomly permuted data did not differ with respect to stimulus-congruence ($2.62 \pm 1.39 \times 10^{-3}$ trials; $\mathrm{T}(4.32 \times 10^3) = 1.85$, $p = 0.06$) but decreased with respect to history-congruence ($2.4 \pm 8.45 \times 10^{-4}$ trials; $\mathrm{T}(4.32 \times 10^3) = -15.26$, $p = 3.11 \times 10^{-51}$) relative to the full model. (**D**) In the likelihood-oscillation-only model, the smoothed probabilities of stimulus-

and history-congruence (sliding windows of ±5 trials) fluctuated as a scale-invariant process with a 1/f power law, i.e., at power densities that were inversely proportional to the frequency (power $\sim 1/f^{\beta}$; stimulus-congruence: $\beta = -0.81 \pm 1.17 \times 10^{-3}$, T$(1.92 \times 10^5) = -688.65$, $p < 2.2 \times 10^{-308}$; history-congruence: $\beta = -0.79 \pm 1.14 \times 10^{-3}$, T$(1.92 \times 10^5) = -698.13$, $p < 2.2 \times 10^{-308}$). (**E**) In the likelihood-oscillation-only model, the distribution of phase shift between fluctuations in simulated stimulus- and history-congruence peaked at half a cycle ($\pi$ denoted by dotted line). In contrast to the full model, the dynamic probabilities of simulated stimulus- and history-congruence were positively correlated ($\beta = 2.7 \times 10^{-3} \pm 7.6 \times 10^{-4}$, T $(2.02 \times 10^6) = 3.55$, $p = 3.8 \times 10^{-4}$). (**F**) In the likelihood-oscillation-only model, the average squared coherence between fluctuations in simulated stimulus- and history-congruence (black dotted line) was reduced in comparison to the full model (T$(3.51 \times 10^3) = -4.56$, $p = 5.27 \times 10^{-6}$) and amounted to $3.43 \pm 1.02 \times 10^{-3}$%. (**G**) Similar to the full bimodal inference model, confidence simulated from the likelihood-oscillation-only model was enhanced for stimulus-congruent choices ($\beta = 0.03 \pm 1.42 \times 10^{-4}$, T$(2.1 \times 10^6) = 191.78$, $p < 2.2 \times 10^{-308}$) and history-congruent choices ($\beta = 9.1 \times 10^{-3} \pm 1.25 \times 10^{-4}$, T$(2.1 \times 10^6) = 72.51$, $p < 2.2 \times 10^{-308}$). (**H**) In the likelihood-oscillation-only model, the positive quadratic relationship between the mode of perceptual processing and confidence was markedly reduced in comparison to the full model ($\beta_2 = 0.34 \pm 0.1$, T$(2.1 \times 10^6) = 3.49$, $p = 4.78 \times 10^{-4}$). The horizontal and vertical dotted lines indicate minimum posterior certainty and the associated mode, respectively.
(TIFF)

**S11 Fig. Reduced control model M3: Only oscillation of the prior.** When simulating data for the *prior-oscillation-only model*, we removed the oscillation from the prior term by setting the amplitude $a_{LLR}$ to 0. Simulated data thus depended only on the participant-wise estimates for hazard rate $H$, amplitude $a_{\psi}$, frequency $f$, phase $p$, and inverse decision temperature $\zeta$. (**A**) Similar to the full model (Figs 1F and 4), simulated perceptual choices were stimulus-congruent in 71.97% ± 0.17% of trials (in red). History-congruent amounted to 52.1% ± 0.11% of trials (in blue). As in the full model, the prior-oscillation-only showed a significant bias toward perceptual history T$(4.32 \times 10^3) = 18.34$, $p = 1.98 \times 10^{-72}$; upper panel). Similarly, history-congruent choices were more frequent at error trials (T$(4.31 \times 10^3) = 12.35$, $p = 1.88 \times 10^{-34}$; lower panel). (**B**) In the prior-oscillation-only model, we did not observe any significant positive autocorrelation of stimulus-congruence, whereas the autocorrelation of history-congruence was preserved. (**C**) In the prior-oscillation-only model, the number of consecutive trials at which true autocorrelation coefficients exceeded the autocorrelation coefficients for randomly permuted data did was decreased with respect to stimulus-congruence relative to the full model $(1.8 \pm 1.01 \times 10^{-3}$ trials; T$(4.31 \times 10^3) = -6.48$, $p = 1.03 \times 10^{-10})$ but did not differ from the full model with respect to history-congruence $(4.25 \pm 1.84 \times 10^{-3}$ trials; T$(4.32 \times 10^3) = 0.07$, $p = 0.95$). (**D**) In the prior-oscillation-only model, the smoothed probabilities of stimulus- and history-congruence (sliding windows of ±5 trials) fluctuated as a scale-invariant process with a 1/f power law, i.e., at power densities that were inversely proportional to the frequency (power $\sim 1/f^{\beta}$; stimulus-congruence: $\beta = -0.78 \pm 1.11 \times 10^{-3}$, T$(1.92 \times 10^5) = -706.62$, $p < 2.2 \times 10^{-308}$; history-congruence: $\beta = -0.83 \pm 1.27 \times 10^{-3}$, T$(1.92 \times 10^5) = -651.6$, $p < 2.2 \times 10^{-308}$). (**E**) In the prior-oscillation-only model, the distribution of phase shift between fluctuations in simulated stimulus- and history-congruence peaked at half a cycle ($\pi$ denoted by dotted line). Similar to the full model, the dynamic probabilities of simulated stimulus- and history-congruence were anticorrelated ($\beta = -0.03 \pm 8.61 \times 10^{-4}$, T$(2.12 \times 10^6) = -34.03$, $p = 8.17 \times 10^{-254}$). (**F**) In the prior-oscillation-only model, the average squared coherence between fluctuations in simulated stimulus- and history-congruence (black dotted line) was reduced in comparison to the full model (T$(3.54 \times 10^3) = -3.22$, $p = 1.28 \times 10^{-3}$) and

amounted to 3.52 ± 1.04×10$^{-3}$%. (**G**) Similar to the full bimodal inference model, confidence simulated from the prior-oscillation-only model was enhanced for stimulus-congruent choices ($\beta$ = 0.02 ± 1.44×10$^{-4}$, T(2.03×10$^6$) = 128.53, $p$ < 2.2×10$^{-308}$) and history-congruent choices ($\beta$ = 0.01 ± 1.26×10$^{-4}$, T(2.03×10$^6$) = 88.24, $p$ < 2.2×10$^{-308}$). (**H**) In contrast to the full bimodal inference model, the prior-oscillation-only model did not yield a positive quadratic relationship between the mode of perceptual processing and confidence ($\beta_2$ = −0.17 ± 0.1, T(2.04×10$^6$) = −1.66, $p$ = 0.1). The horizontal and vertical dotted lines indicate minimum posterior certainty and the associated mode, respectively.
(TIFF)

**S12 Fig. Reduced control model M4: Normative evidence accumulation.** When simulating data for the *normative-evidence-accumulation model*, we removed the oscillation from the likelihood and prior terms by setting the amplitudes $a_{LLR}$ and $a_\psi$ to 0. Simulated data thus depended only on the participant-wise estimates for hazard rate $H$ and inverse decision temperature $\zeta$. (**A**) Similar to the full model (Figs 1F and 4), simulated perceptual choices were stimulus-congruent in 71.97% ± 0.17% of trials (in red). History-congruent amounted to 50.73% ± 0.07% of trials (in blue). As in the full model, the no-oscillation model showed a significant bias toward perceptual history T(4.32×10$^3$) = 9.94, $p$ = 4.88×10$^{-23}$; upper panel). Similarly, history-congruent choices were more frequent at error trials (T(4.31×10$^3$) = 10.59, $p$ = 7.02×10$^{-26}$; lower panel). (**B**) In the normative-evidence-accumulation model, we did not find significant autocorrelations for stimulus-congruence. Likewise, we did not observe any autocorrelation of history-congruence beyond the first 3 consecutive trials. (**C**) In the normative-evidence-accumulation model, the number of consecutive trials at which true autocorrelation coefficients exceeded the autocorrelation coefficients for randomly permuted data decreased with respect to both stimulus-congruence (1.8 ± 1.59×10$^{-3}$ trials; T(4.31×10$^3$) = −5.21, $p$ = 2×10$^{-7}$) and history-congruence (2.18 ± 5.48×10$^{-4}$ trials; T(4.32×10$^3$) = −17.1, $p$ = 1.75×10$^{-63}$) relative to the full model. (**D**) In the normative-evidence-accumulation model, the smoothed probabilities of stimulus- and history-congruence (sliding windows of ±5 trials) fluctuated as a scale-invariant process with a 1/f power law, i.e., at power densities that were inversely proportional to the frequency (power ~ 1/$f^\beta$; stimulus-congruence: $\beta$ = −0.78 ± 1.1×10$^{-3}$, T(1.92×10$^5$) = −706.93, $p$ < 2.2×10$^{-308}$; history-congruence: $\beta$ = −0.79 ± 1.12×10$^{-3}$, T(1.92×10$^5$) = −702.46, $p$ < 2.2×10$^{-308}$). (**E**) In the normative-evidence-accumulation model, the distribution of phase shift between fluctuations in simulated stimulus- and history-congruence peaked at half a cycle ($\pi$ denoted by dotted line). In contrast to the full model, the dynamic probabilities of simulated stimulus- and history-congruence were positively correlated ($\beta$ = 4.3×10$^{-3}$ ± 7.97×10$^{-4}$, T(1.98×10$^6$) = 5.4, $p$ = 6.59×10$^{-8}$). (**F**) In the normative-evidence-accumulation model, the average squared coherence between fluctuations in simulated stimulus- and history-congruence (black dotted line) was reduced in comparison to the full model (T(3.52×10$^3$) = −6.27, $p$ = 3.97×10$^{-10}$) and amounted to 3.26 ± 8.88×10$^{-4}$%. (**G**) Similar to the full bimodal inference model, confidence simulated from the no-oscillation model was enhanced for stimulus-congruent choices ($\beta$ = 0.01 ± 1.05×10$^{-4}$, T(2.1×10$^6$) = 139.17, $p$ < 2.2×10$^{-308}$) and history-congruent choices ($\beta$ = 8.05×10$^{-3}$ ± 9.2×10$^{-5}$, T(2.1×10$^6$) = 85.74, $p$ < 2.2×10$^{-308}$). (**H**) In the normative-evidence-accumulation model, the positive quadratic relationship between the mode of perceptual processing and confidence was markedly reduced in comparison to the full model ($\beta$ = 0.14 ± 0.07, T(2.1×10$^6$) = 1.95, $p$ = 0.05). The horizontal and vertical dotted lines indicate minimum posterior certainty and the associated mode, respectively.
(TIFF)

**S13 Fig. Reduced control model M5: No accumulation of information across trials.** When simulating data for the *no-evidence-accumulation model*, we removed the accumulation of

information across trials by setting the hazard rate $H$ to 0.5. Simulated data thus depended only on the participant-wise estimates for the amplitudes $a_{LLR/\psi}$, frequency $f$, phase $p$, and inverse decision temperature $\zeta$. (**A**) Similar to the full model (Figs 1F and 4), simulated perceptual choices were stimulus-congruent in 72.14% ± 0.17% of trials (in red). History-congruent amounted to 49.89% ± 0.03% of trials (in blue). In contrast to the full model, the no-accumulation model showed a significant bias against perceptual history $T(4.32×10^3) = -3.28$, $p = 1.06×10^{-3}$; upper panel). In contrast to the full model, there was no difference in the frequency of history-congruent choices between correct and error trials ($T(4.31×10^3) = 0.76$, $p = 0.44$; lower panel). (**B**) In the no-evidence-accumulation model, we found no significant autocorrelation of history-congruence beyond the first trial, whereas the autocorrelation of stimulus-congruence was preserved. (**C**) In the no-evidence-accumulation model, the number of consecutive trials at which true autocorrelation coefficients exceeded the autocorrelation coefficients for randomly permuted data increased with respect to stimulus-congruence ($2.83 ± 1.49×10^{-3}$ trials; $T(4.31×10^3) = 3.45$, $p = 5.73×10^{-4}$) and decreased with respect to history-congruence ($1.85 ± 3.49×10^{-4}$ trials; $T(4.32×10^3) = -19.37$, $p = 3.49×10^{-80}$) relative to the full model. (**D**) In the no-evidence-accumulation model, the smoothed probabilities of stimulus- and history-congruence (sliding windows of ±5 trials) fluctuated as a scale-invariant process with a 1/f power law, i.e., at power densities that were inversely proportional to the frequency (power $\sim 1/f^\beta$; stimulus-congruence: $\beta = -0.82 ± 1.2×10^{-3}$, $T(1.92×10^5) = -681.98$, $p < 2.2×10^{-308}$; history-congruence: $\beta = -0.78 ± 1.11×10^{-3}$, $T(1.92×10^5) = -706.57$, $p < 2.2×10^{-308}$). (**E**) In the no-evidence-accumulation model, the distribution of phase shift between fluctuations in simulated stimulus- and history-congruence peaked at half a cycle ($\pi$ denoted by dotted line). In contrast to the full model, the dynamic probabilities of simulated stimulus- and history-congruence were not significantly anticorrelated ($\beta = 6.39×10^{-4} ± 7.22×10^{-4}$, $T(8.89×10^5) = 0.89$, $p = 0.38$). (**F**) In the no-evidence-accumulation model, the average squared coherence between fluctuations in simulated stimulus- and history-congruence (black dotted line) was reduced in comparison to the full model ($T(3.56×10^3) = -9.96$, $p = 4.63×10^{-23}$) and amounted to 2.8 ± 7.29×10^{-4}%. (**G**) Similar to the full bimodal inference model, confidence simulated from the no-evidence-accumulation model was enhanced for stimulus-congruent choices ($\beta = 0.01±9.4×10^{-5}$, $T(2.11×10^6) = 158.1$, $p < 2.2×10^{-308}$). In contrast to the full bimodal inference model, history-congruent choices were not characterized by enhanced confidence ($\beta = 8.78×10^{-5} ± 8.21×10^{-5}$, $T(2.11×10^6) = 1.07$, $p = 0.29$). (**H**) In the no-evidence-accumulation model, the positive quadratic relationship between the mode of perceptual processing and confidence was markedly reduced in comparison to the full model ($\beta_2 = 0.19 ± 0.06$, $T(2.11×10^6) = 3$, $p = 2.69×10^{-3}$). The horizontal and vertical dotted lines indicate minimum posterior certainty and the associated mode, respectively.
(TIFF)

**S14 Fig. Autocorrelation of history-congruence of alternating and repeating biases.** Here, we simulate the autocorrelation of history-congruence in $10^3$ synthetic participants. In the repeating regime (blue), history-congruence fluctuated between 50% and 80% (blue) in interleaved blocks (10 blocks per condition with a random duration between 15 and 30 trials). In the alternation regime (red), history-congruence fluctuated between 50% and 20%. The resulting autocorrelation curves for history-congruence overlap, indicating that our analysis is able to accommodate both repeating and alternating biases.
(TIFF)

**S1 Table. Studies extracted from the Confidence database (downloaded from https://osf. io/s46pr/).**
(PDF)

**S2 Table. Explanation of model parameters.**
(PDF)

## Author Contributions

**Conceptualization:** Veith Weilnhammer, Philipp Sterzer.

**Data curation:** Veith Weilnhammer.

**Formal analysis:** Veith Weilnhammer, Heiner Stuke, Kai Standvoss, Philipp Sterzer.

**Funding acquisition:** Veith Weilnhammer, Philipp Sterzer.

**Investigation:** Veith Weilnhammer.

**Methodology:** Veith Weilnhammer.

**Project administration:** Veith Weilnhammer.

**Resources:** Veith Weilnhammer.

**Software:** Veith Weilnhammer.

**Validation:** Veith Weilnhammer.

**Visualization:** Veith Weilnhammer.

**Writing – original draft:** Veith Weilnhammer, Philipp Sterzer.

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
