## [Editor Report · Decision Letter 0]

2 Jun 2023

Dear Dr Weilnhammer, 

Thank you for submitting your manuscript entitled "Bimodal inference in humans and mice" for consideration as a Research Article by PLOS Biology.

Your manuscript has now been evaluated by the PLOS Biology editorial staff as well as by an academic editor with relevant expertise and I am writing to let you know that we would like to send your submission out for external peer review.

Once your full submission is complete, your paper will undergo a series of checks in preparation for peer review. After your manuscript has passed the checks it will be sent out for review. To provide the metadata for your submission, please Login to Editorial Manager (https://www.editorialmanager.com/pbiology) within two working days, i.e. by Jun 04 2023 11:59PM.

Kind regards,

Christian

Christian Schnell, PhD

Senior Editor

PLOS Biology

cschnell@plos.org

---

## [Decision Letter · Decision Letter 1]

1 Sep 2023

Dear Veith,

Thank you again for your patience while your manuscript "Bimodal inference in humans and mice" was peer-reviewed at PLOS Biology. It has now been evaluated by the PLOS Biology editors, an Academic Editor with relevant expertise, and by several independent reviewers. 

I can only apologise again for the very long delay and thank you for your patience. As I mentioned earlier, the reviewer from a previous submission to another journal had finally agreed to have a look at this manuscript but they weren't able to do so until now and requested an extension. Since another reviewer has, in the meantime, submitted their report, I have discussed the reports with our Academic Editor for this manuscript. We have agreed that the required expertise is covered and that we do not need to keep you waiting even longer and have therefore uninvited the previous reviewer as a reviewer for this submission.

Anyways, In light of the reviews, which you will find at the end of this email, we would like to invite you to revise the work to thoroughly address the reviewers' reports.

As you will see below, the reviewers make a couple of suggestions to improve the manuscript by providing a few additional analyses to rule out alternative explanation. They also emphasise the need to streamline the manuscript such that the key findings are not overshadowed by supplemental findings, and that the manuscript is more approachable for a broader audience.

Given the extent of revision needed, we cannot make a decision about publication until we have seen the revised manuscript and your response to the reviewers' comments. Your revised manuscript is likely to be sent for further evaluation by all or a subset of the reviewers.

**IMPORTANT - SUBMITTING YOUR REVISION**

*Re-submission Checklist*

*Published Peer Review*

*PLOS Data Policy*

*Blot and Gel Data Policy*

Sincerely,

Christian

Christian Schnell, PhD

Senior Editor

PLOS Biology

cschnell@plos.org

REVIEWS:

Reviewer's Responses to Questions

PLOS authors have the option to publish the peer review history of their article (what does this mean?). If published, this will include your full peer review and any attached files.

Reviewer #1: Yes: Karl Friston

Reviewer #2: No

Reviewer #3: No

Reviewer #1: 

Comments to editors

This was an interesting and thought-provoking submission. I note that it is a revision: I am therefore supposing that the authors have already responded to one round of reviewer comments and that you are potentially interested in publishing this work. 

In brief, I think there are many elements of this report that warrant publication; however, there are some parts that are less compelling and could be deferred to a subsequent paper. The paper is far too long and would benefit greatly from being streamlined. Furthermore, some of the modelling is overengineered and is difficult to follow. I have tried to suggest how the authors might improve the presentation of their work in my comments to authors.

I hope that this helps you in your evaluation.

Comments to authors

I enjoyed reading this long but thought-provoking report of fluctuations in the sensitivity to sensory evidence in perceptual decision-making tasks. There were some parts of this report that were compelling and interesting. Other parts were less convincing and difficult to understand.

Overall, this paper is far too long. An analogy that might help here is that a dinner guest is very entertaining for the first hour or so - and then overstays their welcome; until you start wishing they would leave. Another analogy, which came to mind, was that the modelling—and its interpretation—was a bit autistic (i.e., lots of fascinating if questionable detail with a lack of central coherence). 

I think that both issues could be resolved by shortening the paper and removing (or, at least, greatly simplifying) the final simulation studies of metacognition. I try to unpack this suggestion in the following.

Major points

As I understand it, you have used publicly available data on perceptual decision-making to demonstrate slow fluctuations in the tendency to predicate perceptual decisions on the stimuli and on the history of recent decisions. You find scale-free fluctuations in this tendency — that are anti-correlated — and interpret this as fluctuations in the precision afforded sensory evidence, relative to prior beliefs. This interpretation is based upon a model of serial dependencies (parameterised with a hazard function). 

The stimulus and history (i.e., likelihood and prior) sensitivities are anti-correlated and both show scale free behaviour. This is reproduced in men and mice. You then proceed to model this with periodic fluctuations in the precisions or weights applied to the likelihood and prior that are in anti-phase - and then estimate the parameters of the ensuing model. Finally, you then simulate the learning of the hazard parameter — and something called metacognition - to show that periodic fluctuations improve estimates of metacognition (based upon a Rescorla-Wagner model of learning). You motivate this by suggesting that the fluctuations in sensitivity are somehow necessary to elude circular inference and provide better estimates of precision.

Note that I am reading the parameters omega_LLR and omega_ψ as the precision of the likelihood and prior, where the precision of the likelihood is called sensory precision. This contrasts with your use of sensory precision, which seems to be attributed to a metacognitive construct M.

As noted above, all of this is fascinating but there are too many moving parts that do not fit together comfortably. I will list a few examples:

I. If, empirically, the fluctuations in sensitivity are scale-free with a 1/f power law, why did you elect to model fluctuations in precision as a periodic function with one unique timescale (i.e., f).?

II. At present, the estimates of meta-cognition (M) play the role of accumulated estimates of (sensory or prior) precision. Why are these not used in your model of perceptual decisions in Equation 2.

III. Why do you assume that non-specific increases in attention and arousal will increase reaction times? If one has very precise prior beliefs (and is not attending to stimuli), would you not expect a decrease in reaction time?

IV. In the predictive processing literature, attention is thought to correspond to fluctuations in sensory and prior precision. Why did you then consider attention as some additional or unrelated confound?

V. What licences the assumption that "agents depend upon internal confidence signals" in the absence of feedback. And what licences the assumption that internal confidence feedbacks corresponds to "the absolute of the posterior log ratio" (did you mean the log of the posterior ratio)?

VI. I got a bit lost here when you say that "the precision of sensory coding M a function of u_t. This is largely because I couldn't find a definition of u_t.

VII. What licences an application of Rescorla-Wagner to learning the parameters (as in Equation 11) and, learning sensory precision as described by M_T (Equation 13). Are you moving from a Bayesian framework to a reinforcement learning framework?

And so on

I am sure you have answers to these questions - but with each new question the reader is left more and more skeptical that there is a coherent story behind your analyses. It would have been more convincing had you just committed to a Bayesian filter and made your points using one update scheme, under ideal Bayesian observer assumptions. 

Unlike your piecemeal scheme, things like the hierarchical Gaussian filter estimates the sensory and prior decisions explicitly and these estimates underwrite posterior inference. In your scheme, the sensory precision M appears to have no influence on perceptual inference (which is why, presumably you call it metacognition). The problem with this is that your motivation for systematic fluctuations in precision is weakened. This is because improved metacognition does not improve perception — it only improves the perception of perception.

In light of the above, can I suggest that you remove Section 5.8 and use your model in the preceding section to endorse your hypothesis along the following lines:

"In summary, we hypothesized that subjects have certain hyperpriors that are apt for accommodating fluctuations in the predictability of their environment; i.e., people believe that their world is inherently volatile. This means that to be Bayes optimal it is necessary to periodically re-evaluate posterior beliefs about model parameters. One way to do this is to periodically suspend the precision of prior beliefs and increase the precision afforded to sensory evidence that updates (Bayesian) beliefs about model parameters. The empirical evidence above suggests that the timescale of this periodic scheduling of evidence accumulation is scale invariant. This means there exists a timescale of periodic fluctuations in precision over every window or length of perceptual decision-making. In what follows, we model perceptual decisions under a generative model (based upon a hazard function to model historical or serial dependencies) with, a periodic fluctuation in the precision of sensory evidence relative to prior beliefs at a particular timescale. Remarkably—using Bayesian model comparison—we find that a model with fluctuating precisions has much greater evidence, relative to a model in the absence of fluctuating precisions. Furthermore, we were able to quantify the dominant timescale of periodic fluctuations; appropriate for these kinds of paradigm."

Note, again, I am reading your omega_LLR and omega_ψ as precisions and that the periodic modulation is the hyperprior that you are characterizing—and have discovered.

This begs the question as to whether you want to pursue the 1/f story. You refer to this as "noise". However, there is no noise in this setup. I think what you meant was that the fluctuations are scale free, because they evinced a power law. I am sure that there are scale free aspects of these kinds of hyperpriors; however, in the context of your paradigm I wonder whether you should just ignore the scale free aspect and focus on your estimated temporal scale implicit in f. This means you don't have to hand wave about self-organized criticality in the discussion and focus upon your hypothesis.

A final move—to make the paper more focused and digestible—would be to put a lot of your defensive analyses (e.g. about general arousal et cetera) in supplementary material. You have to be careful not to exhaust the reader by putting up a lot of auxiliary material before the important messages in your report.

Minor points

I cannot resist suggesting that you change your title to "Bimodal Inference in Mice and Men"

Please replace "infra-slow fluctuations" with "slow fluctuations". Infra-slow has some colloquial meaning in fMRI studies but not in any scale free context.

Please replace "simulated data" with "simulations" in the abstract. Finally, please replace "robust learning and metacognition in volatile environments" with "enable optimal inference and learning in volatile environments."

Line 50, please replace "about the degree of noise inherent in encoding of sensory information" with "the precision of sensory information relative to prior (Bayesian) beliefs."

Line 125: please replace "a source of error" with "a source of bias"

Line 141: please replace "one 1/f noise" with a scale invariant process with a 1/f power law" (here and throughout) this is not "noise" it is a particular kind of fluctuation. 

Line 178, when you say that the fluctuations may arise due to "changes in level of tonic arousal or on-task attention", I think you need to qualify this. In predictive processing, on-task attention is exactly the modulation of sensory precision, relative to prior precision that you are characterising here. Tonic arousal may be another thing may or may not confound your current results. 

When introducing Equation 2, please make it clear that the omega terms stand in for the precisions afforded to the likelihood (omega_LLR) and prior (omega_ψ) that constitute the log posterior. You can then motivate Equation 6 and 7 as implementing the hyperprior in which the sensory and prior precisions fluctuate at a particular time scale. 

You can also point out that the implicit anti-phase fluctuations are mandated by Bayes optimal formulations in which it is only the relative values of the prior and sensory precision that matter. Bayesian filters these precisions constitute the Kalman gain. You can find a derivation of why this in treatments of the hierarchical Gaussian filter is by Mathys et al.

In your first model simulations, I would make it clear in the main text which parameters you are optimizing's; namely (H, alpha, a_likelihood, a_prior f). Perhaps a little table with a brief description of the meaning of these hyper parameters would be useful?

Please remove Section 5.8. If you do not, you need to explain why — on line 586 - setting a = 0 is appropriate when a = 0, the log posterior in Equation 2 is zero because the precisions (omegas) are zero (by Equations 6 and 7).

I hope that these suggestions help, should any revision be required.

Reviewer #2: Bimodal inference in humans and mice

Veith Weilnhammer, Heiner Stuke, Kai Standvoss, Philipp Sterzer

The authors elucidate whether periodicities in the sensitivity to external information represent an epiphenomenon of limited processing capacity or, alternatively, result from a structured and adaptive mechanism of perceptual inference. Analyzing large datasets of perceptual decision-making in humans and mice, they investigated whether the accuracy of visual perception is constant over time or whether it fluctuates. The authors found significant autocorrelations on the group level and on the level of individual participants, indicating that a stimulus-congruent response in a given trial increased the probability of stimulus-congruent responses in the future. Furthermore, the authors addressed whether observers cycle through periods of enhanced and reduced sensitivity to external information or whether observers rely on internal information in certain phases. This was quantified by whether a response at a given trial was correlated with responses in previous trials. The authors used computational modeling to infer the origin of the different modes (internal vs. external).

Evaluation

This is a very interesting and well-written manuscript, dealing with an important question. The findings are novel and provide an innovative account of interpreting visual perception. I am not an expert in modeling, so I will restrict my comments to the theoretical framework and the experimental approach. I have a few minor questions that I would like the authors to answer or clarify.

Minor questions

1. History congruent perception was defined on the basis of response repetitions. Are we really sure that responses are repeated due to some variant of a perceptual decision process (internal or external) or may arise on the motor-level - independent of a perceptual source? For instance, a response primed by residual activation in the motor system may represent a local effect independent from a general response bias.

2. If indeed, a response repetition is initiated by whatever reasons (non-perceptual), wouldn't this imply that the repeated response is per se more related to previous than to current visual information and would hence signal a reduced sensitivity to current external information? The authors are discussing the option of stereotypically repeated responses in the context of alertness. However, a tendency to repeat responses may arise due to other reasons. For instance, may the motor priming effects mentioned possibly explain faster RTs along with a stronger bias when in internal-mode. 

Reviewer #3: In this paper the authors propose that during perceptual decisions, humans and mice exhibit regular oscillatory fluctuations between an "external" (that places more weight on the perceptual evidence) and an "internal" (that places more weight on historical experiences) mode. In particular, the authors propose a computational scheme in which the influences of history and current stimulus on choice oscillate in anti phase, effectively implementing "bimodal inference". The computational advantages of these scheme as well as its relation to the underlying neurophysiology are discussed.

Overall, the authors make a very interesting proposal about what drives slow fluctuations in perceptual performance during randomised two-alternative choice tasks. This proposal relates changes in accuracy with changes in serial choice biases, which is a timely and synthesising contribution. Furthermore, this proposal is backed by analyses over several human datasets and a large dataset in mice. 

Despite its strong empirical contribution, the paper seems limited by the fact that alternative computational hypotheses are not adequately considered (or at least considered in a systematic way). At the same time, and although the paper is well written, some parts are overly technical. 

Major comments:

1) The authors collapse across various datasets in which different tasks were employed. However, some details on the nature of these different tasks and a discussion on the rationale of collapsing behavioural metrics across them is missing. The authors mention that all tasks involved binary perceptual decisions. In some parts of the manuscript the term "false alarms" is mentioned, indicating a detection protocol. Other terms in the methods section (e.g., "set size") might need further clarification. Importantly, it is not clear how reaction times were calculated in the various tasks and whether some experiments involved free response paradigms while others interrogation/ cued paradigms (in which case RTs can be defined as the latency between the response cue and the response).

2) The key premise that when participants do not rely on the external stimulus they rely more on the previous trial needs to be more clearly (and statistically) contrasted against a null hypothesis. For instance, an null hypothesis could be that when participants place a lower weight on the stimulus they simply choose randomly. It is important to specify a null hypothesis such that the key premise does not appear self-evident or circular. 

3) From a mechanistic (sequential sampling) perspective, several previous papers have examined whether choice history biases influence the starting point or the drift rate of the evidence accumulation process. Under the former formulation, reliance on the evidence vs. reliance on the previous choice will be naturally anti-correlated (the less weight you place on the evidence the more impactful the choice history will be, assuming that the last choice is represented as a starting point bias). This seems to be mapping onto the computational model the authors describe, in which there is a weight on the prior, a weight on the likelihood and the assumption that these weights fluctuate in anti-phase. It is not obvious that this anti-phase relationship needs to be imposed ad-hoc. Or whether it would emerge naturally (using a mechanistic or Bayesian framework). More generally, the authors assert that without an external mechanism prior biases would be impossible to overcome, and this would misfit the data. However, it would be important to a) actually show that the results cannot be explained by a single mechanism in which the anti-phase relationship is emergent rather than ad-hoc, b) relate the current framework with previous mechanistic considerations of serial choice biases.

4) The authors need to unpack their definition of history biases since in previous work biases due to the response or the identity of the stimulus at the previous trial are treated differently. Here, the authors focus on response biases but it is not clear whether they could examine also stimulus-driven history biases (in paradigms where stimulus-response is remapped on each trial).

5) Previous work, which the authors acknowledges in their Discussion (6.5), distinguishes repetitive history biases from alternating biases. For instance, in Braun, Urai & Donner (2018, JoN) participants are split into repetitive or alternating. Shouldn't the authors define the history bias in a similar fashion? The authors point out that attracting and repelling biases operate simultaneously across different timescales. However, this is not warranted given Braun et. al and other similar papers. It is not clear how this more nuanced definition of history bias would alter the conclusions. 

6) The arousal hypothesis seems to be ruled out too easily, merely in the presence of a non-monotonic "state" vs. RT pattern. Arousal can have an inverted U-shaped effect on behavioural performance and recent paper has demonstrated a non-monotonic effect of tonic arousal (baseline pupil) on RTs and accuracy (https://www.biorxiv.org/content/10.1101/2023.07.28.550956.abstract). More generally, the RT and confidence analyses need to be complemented, perhaps by computational modelling using sequential sampling models, as these behavioural metrics have multiple mechanistic mappings (e.g., a fast RT might correspond to high SNR or an impulsive decisions driven by a starting point bias).

7) In several analysis the authors present an effect and then show that this effecs persists when key variables/ design aspects are also taken into account (see an example at around line 70). It makes more sense to present only one single analysis in which these key variables are controlled for. Results cannot be interpreted if they are spurious factors driving them so it is not clear why some of the results are presented in two versions ("uncontrolled" and "controlled" analyses).

8) The central empirical finding is potentially important but is currently shadowed by more speculative sections/ discussions. For instance, the section on the adaptive merits of the computational model is relatively weaker compared to the empirical results. In particular, the model is simulated without feedback (whereas most experiments employ trial by trial feedback) and does not outperform the baseline model in accuracy but in other secondary metrics. 

Minor comments:

-- The amount of statistical analysis and results is often overwhelming. The authors could streamline the presentation better such that the main result is brought to the foreground. Currently the manuscript resembles a technical report. 

-- Some typos or omissions may alter the meaning in various places. Indicatively, in lines 273, 439, 649.

---

## [Decision Letter · Decision Letter 2]

17 Oct 2023

Dear Veith,

Thank you for your patience while we considered your revised manuscript "Bimodal inference in humans and mice" for publication as a Research Article at PLOS Biology. This revised version of your manuscript has been evaluated by the PLOS Biology editors, the Academic Editor .

Based on the reviews and on our Academic Editor's assessment of your revision, we are likely to accept this manuscript for publication, provided you satisfactorily address the following data and other policy-related requests.

* We would like to suggest a different title to improve readability for our broad readership: 

Sensory processing in humans and mice fluctuates between external and internal modes

* Please provide a link to the funding agencies in the Financial disclosure statement. 

* Thank you for making the data, code, and manuscript data available on github. Could please also provide a persistent identifier (DOI) for this repository. You can obtain a DOI and link it to your github repository for example with zenodo.

* DATA POLICY:

2B, 2C, 2G, 3B, 3C, 3G, 4C, 4B, and 4G.

We expect to receive your revised manuscript within two weeks. 

*Published Peer Review History*

*Press*

Sincerely,

Christian

Christian Schnell, PhD

Senior Editor,

cschnell@plos.org,

PLOS Biology

Reviewer remarks:

Reviewer #1 (Karl Friston): Many thanks for attending to my previous suggestions and the substantive revisions. I think this streamlined report is much more compelling and easier to navigate. Congratulations on a thoughtful piece of work.

Reviewer #2: The authors answered all my questions. I have no further. 

Reviewer #3 (Konstantinos Tsetsos): The authors have done an excellent job in addressing my comments by performing extensive additional model comparisons and statistical analyses. They have also shortened the main text and streamlined the results in a less technical manner. I am confident that the paper will be accessible and of interest to a broad audience.

---

## [Editor Report · Decision Letter 3]

23 Oct 2023

Dear Veith,

Thank you for your patience while we considered your revised manuscript "Sensory processing in humans and mice fluctuates between external and internal modes" for publication as a Research Article at PLOS Biology.

Thank you for addressing and implementing the requests from our previous email. One item, however, we could not find: The individual datapoints that underlie the data summarized in the figures and results of your paper. For more information, please also see this editorial: http://dx.doi.org/10.1371/journal.pbio.1001797

2B, 2C, 2G, 3B, 3C, 3G, 4C, 4B, and 4G.

We expect to receive your revised manuscript within two weeks. 

*Published Peer Review History*

*Press*

Sincerely,

Christian

Christian Schnell, PhD

Senior Editor,

cschnell@plos.org,

PLOS Biology

---

## [Editor Report · Decision Letter 4]

30 Oct 2023

Dear Veith,

Thank you for the submission of your revised Research Article "Sensory processing in humans and mice fluctuates between external and internal modes" for publication in PLOS Biology. On behalf of my colleagues and the Academic Editor, Thorsten Kahnt, I am pleased to say that we can in principle accept your manuscript for publication, provided you address any remaining formatting and reporting issues. These will be detailed in an email you should receive within 2-3 business days from our colleagues in the journal operations team; no action is required from you until then. Please note that we will not be able to formally accept your manuscript and schedule it for publication until you have completed any requested changes.

PRESS

Sincerely, 

Christian

Christian Schnell, PhD, PhD

Senior Editor

PLOS Biology

cschnell@plos.org